# Adversarial Multiclass Classification:
# A Risk Minimization Perspective

**Rizal Fathony**     **Anqi Liu**     **Kaiser Asif**     **Brian D. Ziebart**
Department of Computer Science
University of Illinois at Chicago
Chicago, IL 60607
{rfatho2, aliu33, kasif2, bziebart}@uic.edu

## Abstract

Recently proposed adversarial classification methods have shown promising results for cost sensitive and multivariate losses. In contrast with empirical risk minimization (ERM) methods, which use convex surrogate losses to approximate the desired non-convex target loss function, adversarial methods minimize non-convex losses by treating the properties of the training data as being uncertain and worst case within a minimax game. Despite this difference in formulation, we recast adversarial classification under zero-one loss as an ERM method with a novel prescribed loss function. We demonstrate a number of theoretical and practical advantages over the very closely related hinge loss ERM methods. This establishes adversarial classification under the zero-one loss as a method that fills the long standing gap in multiclass hinge loss classification, simultaneously guaranteeing Fisher consistency and universal consistency, while also providing dual parameter sparsity and high accuracy predictions in practice.

## 1   Introduction

A common goal for standard classification problems in machine learning is to find a classifier that minimizes the zero-one loss. Since directly minimizing this loss over training data via empirical risk minimization (ERM) [1] is generally NP-hard [2], convex surrogate losses are employed to approximate the zero-one loss. For example, the logarithmic loss is minimized by the logistic regression classifier [3] and the hinge loss is minimized by the support vector machine (SVM) [4, 5]. Both are Fisher consistent [6, 7] and universally consistent [8, 9] for binary classification, meaning they minimize the zero-one loss and are Bayes-optimal classifiers when they learn from any true distribution of data using a rich feature representation. SVMs provide the additional advantage of dual parameter sparsity so that when combined with kernel methods, extremely rich feature representations can be efficiently considered. Unfortunately, generalizing the hinge loss to classification tasks with more than two labels is challenging and existing multiclass convex surrogates [10–12] tend to lose their consistency guarantees [13–15] or produce low accuracy predictions in practice [15].

Adversarial classification [16, 17] uses a different approach to tackle non-convex losses like the zero-one loss. Instead of approximating the desired loss function and evaluating over the training data, it adversarially approximates the available training data within a minimax game formulation with game payoffs defined by the desired (zero-one) loss function [18, 19]. This provides promising empirical results for cost-sensitive losses [16] and multivariate losses such as the F-measure and the precision-at-k [17]. Conceptually, parameter optimization for the adversarial method forces the adversary to "behave like" certain properties of the training data sample, making labels easier to predict within the minimax prediction game. However, a key bottleneck for these methods has been

their reliance on zero-sum game solvers for inference, which are computationally expensive relative to inference in other prediction methods, such as SVMs.

In this paper, we recast adversarial prediction from an empirical risk minimization perspective by analyzing the Nash equilibrium value of adversarial zero-one classification games to define a new multiclass loss[1]. This enables us to demonstrate that zero-one adversarial classification fills the long standing gap in ERM-based multiclass classification by simultaneously: (1) guaranteeing Fisher consistency and universal consistency; (2) enabling computational efficiency via the kernel trick and dual parameter sparsity; and (3) providing competitive performance in practice. This reformulation also provides significant computational efficiency improvements compared to previous adversarial classification training methods [16].

## 2 Background and Related Work

### 2.1 Multiclass SVM generalizations

The multiclass support vector machine (SVM) seeks class-based potentials $f_y(\mathbf{x}_i)$ for each input vector $\mathbf{x} \in \mathcal{X}$ and class $y \in \mathcal{Y}$ so that the discriminant function, $\hat{y}_{\mathbf{f}}(\mathbf{x}_i) = \operatorname{argmax}_y f_y(\mathbf{x}_i)$, minimizes misclassification errors, $\text{loss}_{\mathbf{f}}(\mathbf{x}_i, y_i) = I(y_i \neq \hat{y}_{\mathbf{f}}(\mathbf{x}_i))$. Unfortunately, empirical risk minimization (ERM), $\min_{\mathbf{f}} \mathbb{E}_{\tilde{P}(\mathbf{x},y)} [\text{loss}_{\mathbf{f}}(\mathbf{X}, Y)]$, for the zero-one loss is NP-hard once the set of potentials is (parametrically) restricted (e.g., as a linear function of input features) [2]. Instead, a hinge loss approximation is employed by the SVM. In the binary setting, $y_i \in \{-1, +1\}$, where the potential of one class can be set to zero ($f_{-1} = 0$) with no loss in generality, the hinge loss is defined as $[1 - y_i f_{+1}(\mathbf{x}_i)]_+$, with the compact definition $[g(.)]_+ \triangleq \max(0, g(.))$. Binary SVM, which is an empirical risk minimizer using the hinge loss with $L_2$ regularization,

$$\min_{\mathbf{f}_\theta} \mathbb{E}_{\tilde{P}(\mathbf{x},y)} [\text{loss}_{\mathbf{f}_\theta}(\mathbf{X}, Y)] + \tfrac{\lambda}{2}||\theta||_2^2, \tag{1}$$

provides strong theoretical guarantees (Fisher consistency and universal consistency) [8, 21] and computational efficiency [1].

Many methods have been proposed to generalize SVM to the multiclass setting. Apart from the one-vs-all and one-vs-one decomposed formulations [22], there are three main joint formulations: the WW model by Weston et al. [11], which incorporates the sum of hinge losses for all alternative labels, $\text{loss}_{\text{WW}}(\mathbf{x}_i, y_i) = \sum_{j \neq y_i} [1 - (f_{y_i}(\mathbf{x}_i) - f_j(\mathbf{x}_i))]_+$; the CS model by Crammer and Singer [10], which uses the hinge loss of only the largest alternative label, $\text{loss}_{\text{CS}}(\mathbf{x}_i, y_i) = \max_{j \neq y_i} [1 - (f_{y_i}(\mathbf{x}_i) - f_j(\mathbf{x}_i))]_+$; and the LLW model by Lee et al. [12], which employs an absolute hinge loss, $\text{loss}_{\text{LLW}}(\mathbf{x}_i, y_i) = \sum_{j \neq y_i} [1 + f_j(\mathbf{x}_i)]_+$, and a constraint that $\sum_j f_j(\mathbf{x}_i) = 0$. The former two models (CS and WW) both utilize the pairwise class-based potential differences $f_{y_i}(\mathbf{x}_i) - f_j(\mathbf{x}_i)$ and are therefore categorized as relative margin methods. LLW, on the other hand, is an absolute margin method that only relates to $f_j(\mathbf{x}_i)$[15]. Fisher consistency, or Bayes consistency [7, 13] guarantees that minimization of a surrogate loss for the true distribution provides the Bayes-optimal classifier, i.e., minimizes the zero-one loss. If given any possible distribution of data, a classifier is Bayes-optimal, it is called universally consistent. Of these, only the LLW method is Fisher consistent and universally consistent [12–14]. However, as pointed out by Doğan et al. [15], LLW's use of an absolute margin in the loss (rather than the relative margin of WW and CS) often causes it to perform poorly for datasets with low dimensional feature spaces. From the opposite direction, the requirements for Fisher consistency have been well-characterized [13], yet this has not led to a multiclass classifier that is both Fisher consistent and performs well in practice.

### 2.2 Adversarial prediction games

Building on a variety of diverse formulations for adversarial prediction [23–26], Asif et al. [16] proposed an adversarial game formulation for multiclass classification with cost-sensitive loss functions. Under this formulation, the empirical training data is replaced by an adversarially chosen conditional label distribution $\check{P}(\check{y}|\mathbf{x})$ that must closely approximate the training data, but otherwise

seeks to maximize expected loss, while an estimator player $\hat{P}(\hat{y}|\mathbf{x})$ seeks to minimize expected loss. For the zero-one loss, the prediction game is:

$$\min_{\hat{P}} \max_{\check{P}:\mathbb{E}_{P(\mathbf{x})\check{P}(\check{y}|\mathbf{x})}[\phi(\mathbf{X},\check{Y})]=\tilde{\phi}} \mathbb{E}_{\tilde{P}(\mathbf{x})\hat{P}(\hat{y}|\mathbf{x})\check{P}(\check{y}|\mathbf{x})}\left[I(\hat{Y} \neq \check{Y})\right]. \tag{2}$$

The vector of feature moments, $\tilde{\phi} = \mathbb{E}_{\tilde{P}(\mathbf{x},y)}[\phi(\mathbf{X},Y)]$, is measured from sample training data. Using minimax and strong Lagrangian duality, the optimization of Eq. (2) reduces to minimizing the equilibrium game values of a new set of zero-sum games characterized by matrix $\mathbf{L}'_{\mathbf{x}_i,\theta}$:

$$\min_{\theta} \sum_i \max_{\check{\mathbf{p}}} \min_{\hat{\mathbf{p}}} \hat{\mathbf{p}}_{\mathbf{x_i}}^T \mathbf{L}'_{\mathbf{x}_i,\theta} \check{\mathbf{p}}_{\mathbf{x_i}}; \quad \mathbf{L}'_{\mathbf{x}_i,\theta} = \begin{bmatrix} \psi_{1,y_i}(\mathbf{x}_i) & \cdots & \psi_{|\mathcal{Y}|,y_i}(\mathbf{x}_i)+1 \\ \vdots & \ddots & \vdots \\ \psi_{1,y_i}(\mathbf{x}_i)+1 & \cdots & \psi_{|\mathcal{Y}|,y_i}(\mathbf{x}_i) \end{bmatrix}; \tag{3}$$

where $\theta$ is a vector of Lagrangian model parameters, $\hat{\mathbf{p}}_{\mathbf{x}_i}$ is a vector representation of the conditional label distribution, $\hat{P}(\hat{Y} = k|\mathbf{x}_i)$, i.e. $\hat{\mathbf{p}}_{\mathbf{x}_i} = [\hat{P}(\hat{Y} = 1|\mathbf{x}_i) \ \hat{P}(\hat{Y} = 2|\mathbf{x}_i) \ \ldots]^T$, and similarly for $\check{\mathbf{p}}_{\mathbf{x}_i}$. The matrix $\mathbf{L}'_{\mathbf{x}_i,\theta}$ is a zero-sum game matrix for each example, with $\psi_{j,y_i}(\mathbf{x}_i) = f_j(\mathbf{x}_i) - f_{y_i}(\mathbf{x}_i) = \theta^{\mathrm{T}} (\phi(\mathbf{x}_i, j) - \phi(\mathbf{x}_i, y_i))$. This optimization problem (Eq. (3)) is convex in $\theta$ and the inner zero-sum game can be solved using linear programming [16].

## 3 Risk Minimization Perspective of Adversarial Multiclass Classification

### 3.1 Nash equilibrium game value

Despite the differences in formulation between adversarial loss minimization and empirical risk minimization, we now recast the zero-one loss adversarial game as the solution to an empirical risk minimization problem. Theorem 1 defines the loss function that provides this equivalence by considering all possible combinations of the adversary's label assignments with non-zero probability in the Nash equilibrium of the game.[2]

**Theorem 1.** *The model parameters $\theta$ for multiclass zero-one adversarial classification are equivalently obtained from empirical risk minimization under the adversarial zero-one loss function:*

$$AL_{\mathbf{f}}^{0\text{-}1}(\mathbf{x}_i, y_i) = \max_{\mathcal{S} \subseteq \{1,\ldots,|\mathcal{Y}|\}, \ \mathcal{S} \neq \emptyset} \frac{\sum_{j\in\mathcal{S}} \psi_{j,y_i}(\mathbf{x}_i) + |\mathcal{S}| - 1}{|\mathcal{S}|}, \tag{4}$$

*where $\mathcal{S}$ is any non-empty member of the powerset of classes $\{1, 2, \ldots, |\mathcal{Y}|\}$.*

Thus, $AL^{0\text{-}1}$ is the maximum value over $2^{|\mathcal{Y}|} - 1$ linear hyperplanes. For binary prediction tasks, there are three linear hyperplanes: $\psi_{1,y}(\mathbf{x}), \psi_{2,y}(\mathbf{x})$ and $\frac{\psi_{1,y}(\mathbf{x})+\psi_{2,y}(\mathbf{x})+1}{2}$. Figure 1 shows the loss function in potential difference spaces $\psi$ when the true label is $y = 1$. Note that $AL^{0\text{-}1}$ combines two hinge functions at $\psi_{2,y}(\mathbf{x}) = -1$ and $\psi_{2,y}(\mathbf{x}) = 1$, rather than SVM's single hinge at $\psi_{1,y}(\mathbf{x}) = -1$. This difference from the hinge loss corresponds to the loss that is realized by randomizing label predictions.[3] For three classes, the loss function has seven facets as shown in Figure 2a. Figures 2a, 2b, and 2c show the similarities and differences between $AL^{0\text{-}1}$ and the multiclass SVM surrogate losses based on class potential differences. Note that $AL^{0\text{-}1}$ is also a relative margin loss function that utilizes the pairwise potential difference $\psi_{j,y}(\mathbf{x})$.

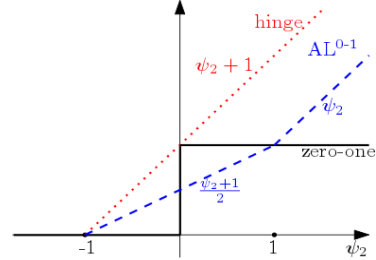

Figure 1: $AL^{0\text{-}1}$ evaluated over the space of potential differences $(\psi_{j,y}(\mathbf{x}) = f_j(\mathbf{x}) - f_y(\mathbf{x})$; and $\psi_{j,j}(\mathbf{x}) = 0)$ for binary prediction tasks when the true label is $y = 1$.

### 3.2 Consistency properties

Fisher consistency is a desirable property for a surrogate loss function that guarantees its minimizer, given the true distribution, $P(x, y)$, will yield the Bayes optimal decision boundary [13, 14]. For

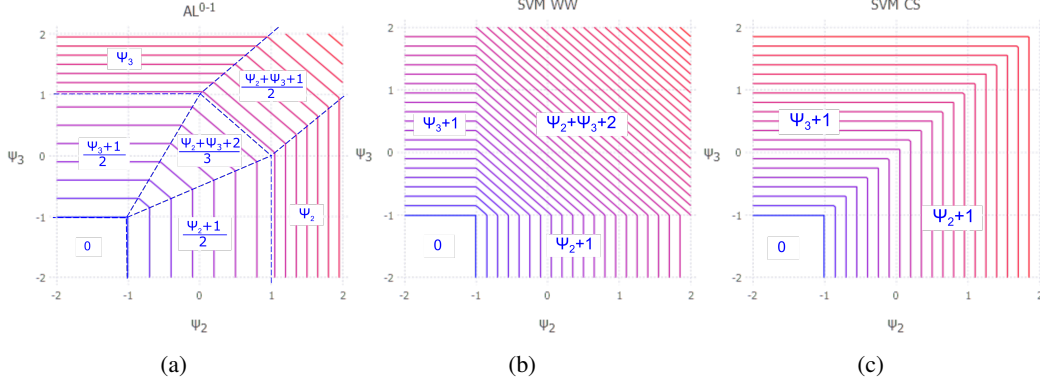

Figure 2: Loss function contour plots over the space of potential differences for the prediction task with three classes when the true label is $y = 1$ under $\text{AL}^{0\text{-}1}$ (a), the WW loss (b), and the CS loss (c). (Note that $\psi_i$ in the plots refers to $\psi_{j,y}(\mathbf{x}) = f_j(\mathbf{x}) - f_y(\mathbf{x})$; and $\psi_{j,j}(\mathbf{x}) = 0$.)

multiclass zero-one loss, given that we know $P_j(\mathbf{x}) \triangleq P(Y = j|\mathbf{x})$, Fisher consistency requires that $\operatorname{argmax}_j f_j^*(\mathbf{x}) \subseteq \operatorname{argmax}_j P_j(\mathbf{x})$, where $\mathbf{f}^*(\mathbf{x}) = [f_1^*(\mathbf{x}), \ldots, f_{|\mathcal{Y}|}^*(\mathbf{x})]^{\mathrm{T}}$ is the minimizer of $\mathbb{E}\left[\text{loss}_{\mathbf{f}}(\mathbf{X}, Y)|\mathbf{X} = \mathbf{x}\right]$. Since any constant can be added to all $f_j^*(\mathbf{x})$ while keeping $\operatorname{argmax}_j f_j^*(\mathbf{x})$ the same, we employ a sum-to-zero constraint, $\sum_{j=1}^{|\mathcal{Y}|} f_j(\mathbf{x}) = 0$, to remove redundant solutions. We establish an important property of the minimizer for $\text{AL}^{0\text{-}1}$ in the following theorem.

**Theorem 2.** *The loss for the minimizer $\mathbf{f}^*$ of $\mathbb{E}\left[AL_{\mathbf{f}}^{0\text{-}1}(\mathbf{X}, Y)|\mathbf{X} = \mathbf{x}\right]$ resides on the hyperplane defined (in Eq. 4) by the complete set of labels, $\mathcal{S} = \{1, \ldots, |\mathcal{Y}|\}$.*

As an illustration for the case of three classes (Figure 2a), the area described in the theorem above corresponds to the region in the middle where the hyperplane that supports $\text{AL}^{0\text{-}1}$ is $\frac{\psi_{1,y}(\mathbf{x}) + \psi_{2,y}(\mathbf{x}) + \psi_{3,y}(\mathbf{x}) + 2}{3}$, and, equivalently, where $-\frac{1}{|\mathcal{Y}|} \leq f_j(\mathbf{x}) \leq \frac{|\mathcal{Y}|-1}{|\mathcal{Y}|}, \forall j \in \{1, \ldots, |\mathcal{Y}|\}$ with a constraint that $\sum_j f_j(\mathbf{x}) = 0$. Based on this restriction, we focus on the minimization of $\mathbb{E}\left[\text{AL}_{\mathbf{f}}^{0\text{-}1}(\mathbf{X}, Y)|\mathbf{X} = \mathbf{x}\right]$ subject to $-\frac{1}{|\mathcal{Y}|} \leq f_j(\mathbf{x}) \leq \frac{|\mathcal{Y}|-1}{|\mathcal{Y}|}, \forall j \in \{1, \ldots, |\mathcal{Y}|\}$ and the sum of potentials equal to zero. This minimization reduces to the following optimization:

$$\max_{\mathbf{f}} \sum_{y=1}^{|\mathcal{Y}|} P_y(\mathbf{x})f_y(\mathbf{x}) \text{ subject to: } -\frac{1}{|\mathcal{Y}|} \leq f_j(\mathbf{x}) \leq \frac{|\mathcal{Y}|-1}{|\mathcal{Y}|} \quad j \in \{1, \ldots, |\mathcal{Y}|\}; \quad \sum_{j=1}^{|\mathcal{Y}|} f_j(\mathbf{x}) = 0.$$

The solution for this maximization (a linear program) satisfies $f_j^*(\mathbf{x}) = \frac{|\mathcal{Y}|-1}{|\mathcal{Y}|}$ if $j = \operatorname{argmax}_j P_j(\mathbf{x})$, and $-\frac{1}{|\mathcal{Y}|}$ otherwise, which therefore implies the Fisher consistency theorem.

**Theorem 3.** *The adversarial zero-one loss, $AL^{0\text{-}1}$, from Eq.* (4) *is Fisher consistent.*

Theorem 3 implies that $\text{AL}^{0\text{-}1}$ (Eq. (4)) is classification calibrated, which indicates minimization of that loss for all distributions on $\mathcal{X} \times \mathcal{Y}$ also minimizes the zero-one loss [21, 13]. As proven in general by Steinwart and Christmann [2], Micchelli et al. [27], since $\text{AL}^{0\text{-}1}$ (Eq.(4)) is a Lipschitz loss with constant 1, the adversarial multiclass classifier is universally consistent under the conditions specified in Corollary 1.

**Corollary 1.** *Given a universal kernel and regularization parameter $\lambda$ in Eq.* (1) *tending to zero slower than $\frac{1}{n}$, the adversarial multiclass classifier is also universally consistent.*

### 3.3 Optimization

In the learning process for adversarial classification, Asif et al. [16] requires a linear program to be solved that finds the Nash equilibrium game value and strategy for every training data point in each gradient update. This requirement is computationally burdensome compared to multiclass SVMs, which must simply find potential-maximizing labels. We propose two approaches with improved

efficiency by leveraging an oracle for finding the maximization inside $AL^{0\text{-}1}$ and Lagrange duality in the quadratic programming formulation.

### 3.3.1 Primal optimization using stochastic sub-gradient descent

The sub-gradient in the empirical risk minimization of $AL^{0\text{-}1}$ includes the mean of feature differences, $\frac{1}{|R|} \sum_{j \in R} [\phi(\mathbf{x}_i, j) - \phi(\mathbf{x}_i, y_i)]$, where $R$ is the set that maximizes $AL^{0\text{-}1}$. The set $R$ is computed by the oracle using a greedy algorithm. Given $\theta$ and a sample $(\mathbf{x}_i, y_i)$, the algorithm calculates all potentials $\psi_{j,y_i}(\mathbf{x}_i)$ for each label $j \in \{1, \dots, |\mathcal{Y}|\}$ and sorts them in non-increasing order. Starting with the empty set $R = \emptyset$, it then adds labels to $R$ in sorted order until adding a label would decrease the value of $\frac{\sum_{j \in R} \psi_{j,y_i}(\mathbf{x}_i) + |R| - 1}{|R|}$.

**Theorem 4.** *The proposed greedy algorithm used by the oracle is optimal.*

### 3.3.2 Dual optimization

In the next subsections, we focus on the dual optimization technique as it enables us to establish convergence guarantees. We re-formulate the learning algorithm (with $L_2$ regularization) as a constrained quadratic program (QP) with $\xi_i$ specifying the amount of $AL^{0\text{-}1}$ incurred by each of the $n$ training examples:

$$\min_{\theta} \frac{1}{2}\|\theta\|^2 + C \sum_{i=1}^{n} \xi_i \quad \text{subject to:} \quad \xi_i \geq \Delta_{i,k} \quad \forall i \in \{1, \dots n\} k \in \{1, \dots, 2^{|\mathcal{Y}|} - 1\}, \quad (5)$$

where we denote each of the $2^{|\mathcal{Y}|} - 1$ possible constraints for example $i$ corresponding to non-empty elements of the label powerset as $\Delta_{i,k}$ (e.g., $\Delta_{i,1} = \psi_{1,y_i}(\mathbf{x}_i)$, and $\Delta_{i,2^{|\mathcal{Y}|}-1} = \frac{\sum_{j \in \mathcal{Y}} \psi_{j,y_i}(\mathbf{x}_i) + |\mathcal{Y}| - 1}{|\mathcal{Y}|}$). Note also that non-negativity for $\xi_i$ is enforced since $\Delta_{i,y_i} = \psi_{y_i,y_i}(\mathbf{x}_i) = 0$.

**Theorem 5.** *Let $\Lambda_{i,k}$ be the partial derivative of $\Delta_{i,k}$ with respect to $\theta$, i.e., $\Lambda_{i,k} = \frac{d\Delta_{i,k}}{d\theta}$ and $\nu_{i,k}$ is the constant part of $\Delta_{i,k}$ (for example if $\Delta_{i,k} = \frac{\psi_{1,y_i}(\mathbf{x}_i) + \psi_{3,y_i}(\mathbf{x}_i) + \psi_{4,y_i}(\mathbf{x}_i) + 2}{3}$, then $\nu_{i,k} = \frac{2}{3}$), then the corresponding dual optimization for the primal minimization (Eq. 5) is:*

$$\max_{\boldsymbol{\alpha}} \sum_{i=1}^{n} \sum_{k=1}^{2^{|\mathcal{Y}|}-1} \nu_{i,k} \, \alpha_{i,k} - \frac{1}{2} \sum_{i,j=1}^{m} \sum_{k,l=1}^{2^{|\mathcal{Y}|}-1} \alpha_{i,k} \alpha_{j,l} \left[ \Lambda_{i,k} \cdot \Lambda_{j,l} \right] \quad (6)$$

$$\text{subject to:} \quad \alpha_{i,k} \geq 0, \quad \sum_{k=1}^{2^{|\mathcal{Y}|}-1} \alpha_{i,k} = C, \; i \in \{1, \dots, n\}, \, k \in \{1, \dots, 2^{|\mathcal{Y}|} - 1\},$$

*where $\alpha_{i,k}$ is the dual variable for the $k$-th constraint of the $i$-th sample.*

Note that the dual formulation above only depends on the dot product of two constraints' partial derivatives (with respect to $\theta$) and the constant part of the constraints. The original primal variable $\theta$ can be recovered from the dual variables using the formula: $\theta = -\sum_{i=1}^{n} \sum_{k=1}^{2^{|\mathcal{Y}|}-1} \alpha_{i,k} \Lambda_{i,k}$. Given a new datapoint $\mathbf{x}$, de-randomized predictions are obtained from $\text{argmax}_j f_j(\mathbf{x}) = \text{argmax}_j \theta^{\mathrm{T}} \phi(\mathbf{x}, j)$.

### 3.3.3 Efficiently incorporating rich feature spaces using kernelization

Considering large feature spaces is important for developing an expressive classifier that can learn from large amounts of training data. Indeed, Fisher consistency requires such feature spaces for its guarantees to be meaningful. However, naïvely projecting from the original input space, $\mathbf{x}_i$, to richer (or possibly infinite) feature spaces $\omega(\mathbf{x}_i)$, can be computationally burdensome. Kernel methods enable this feature expansion by allowing the dot products of certain feature functions to be computed implicitly, i.e., $K(\mathbf{x}_i, \mathbf{x}_j) = \omega(\mathbf{x}_i) \cdot \omega(\mathbf{x}_j)$. Since our dual formulation only depends on dot products, we employ kernel methods to incorporate rich feature spaces into our formulation as stated in the following theorem.

**Theorem 6.** *Let $\mathcal{X}$ be the input space and $K$ be a positive definite real valued kernel on $\mathcal{X} \times \mathcal{X}$ with a mapping function $\omega(\mathbf{x}) : \mathcal{X} \rightarrow \mathcal{H}$ that maps the input space $\mathcal{X}$ to a reproducing kernel Hilbert*

space $\mathcal{H}$. Then all the values in the dual optimization of Eq. (6) needed to operate in the Hilbert space $\mathcal{H}$ can be computed in terms of the kernel function $K(\mathbf{x}_i, \mathbf{x}_j)$ as:

$$\Lambda_{i,k} \cdot \Lambda_{j,l} = c_{(i,k),(j,l)}\, K(\mathbf{x}_i, \mathbf{x}_j), \quad \Delta_{i,k} = -\sum_{j=1}^{n}\sum_{l=1}^{2^{|\mathcal{Y}|}-1} \alpha_{j,l}\, c_{(j,l),(i,k)}\, K(\mathbf{x}_j, \mathbf{x}_i) + \nu_{i,k}, \quad (7)$$

$$f_m(\mathbf{x}_i) = -\sum_{j=1}^{n}\sum_{l=1}^{2^{|\mathcal{Y}|}-1} \alpha_{j,l} \left[ \left( \frac{\mathbf{1}(m \in R_{j,l})}{|R_{j,l}|} - \mathbf{1}(m = y_j) \right) K(\mathbf{x}_j, \mathbf{x}_i) \right], \quad (8)$$

where $c_{(i,k),(j,l)} = \sum\limits_{m=1}^{|\mathcal{Y}|} \left( \dfrac{\mathbf{1}(m \in R_{i,k})}{|R_{i,k}|} - \mathbf{1}(m = y_i) \right) \left( \dfrac{\mathbf{1}(m \in R_{j,l})}{|R_{j,l}|} - \mathbf{1}(m = y_j) \right),$

and $R_{i,k}$ is the set of labels included in the constraint $\Delta_{i,k}$ (for example if $\Delta_{i,k} = \frac{\psi_{1,y_i}(\mathbf{x}_i) + \psi_{3,y_i}(\mathbf{x}_i) + \psi_{4,y_i}(\mathbf{x}_i) + 2}{3}$, then $R_{i,k} = \{1,3,4\}$), the function $\mathbf{1}(j = y_i)$ returns 1 if $j = y_i$ or 0 otherwise, and the function $\mathbf{1}(j \in R_{i,k})$ returns 1 if $j$ is a member of set $R_{i,k}$ or 0 otherwise.

### 3.3.4 Efficient optimization using constraint generation

The number of constraints in the QP formulation above grows exponentially with the number of classes: $\mathcal{O}(2^{|\mathcal{Y}|})$. This prevents the naïve formulation from being efficient for large multi-class problems. We employ a constraint generation method to efficiently solve the dual quadratic programming formulation that is similar to those used for extending the SVM to multivariate loss functions [28] and structured prediction settings [29].

---

**Algorithm 1** Constraint generation method

---

**Require:** Training data $(\mathbf{x}_1, y_1), \ldots (\mathbf{x}_n, y_n), C, \epsilon$
 1: $\theta \leftarrow \mathbf{0}$
 2: $A_i^* \leftarrow \{\Delta_{i,k} | \Delta_{i,k} = \psi_{y_i,y_i}(\mathbf{x}_i)\}\ \forall i = 1, \ldots, n$         $\triangleright$ Actual label enforces non-negativity
 3: **repeat**
 4:      **for** $i \leftarrow 1, n$ **do**
 5:          $a \leftarrow \arg\max_{k | \Delta_{i,k} \in A_i} \Delta_{i,k}$         $\triangleright$ Find the most violated constraint
 6:          $\xi_i \leftarrow \max_{k | \Delta_{i,k} \in A_i^*} \Delta_{i,k}$         $\triangleright$ Compute the example's current loss estimate
 7:          **if** $\Delta_{i,a} > \xi_i + \epsilon$ **then**
 8:             $A_i^* \leftarrow A_i^* \cup \{\Delta_{i,a}\}$         $\triangleright$ Add it to the enforced constraints set
 9:             $\boldsymbol{\alpha} \leftarrow$ Optimize dual over $A^* = \cup_i A_i^*$
10:             Compute $\theta$ from $\boldsymbol{\alpha}$: $\theta = -\sum_{i=1}^n \sum_{k | \Delta_{i,k} \in A_i^*} \alpha_{i,k} \Lambda_{i,k}$
11:          **end if**
12:      **end for**
13: **until** no $A_i^*$ has changed in the iteration

---

Algorithm 1 incrementally expands the set of enforced constraints, $A_i^*$, until no remaining constraint from the set of all $2^{|\mathcal{Y}|} - 1$ constraints (in $A_i$) is violated by more than $\epsilon$. To obtain the most violated constraint, we use the greedy algorithm described in the primal optimization. The constraint generation algorithm's stopping criterion ensures that a solution close to the optimal is returned (violating no constraint by more than $\epsilon$). Theorem 7 provides a polynomial run time convergence bounds for the Algorithm 1.

**Theorem 7.** *For any $\epsilon > 0$ and training dataset $\{(\mathbf{x}_1, y_1), \ldots, (\mathbf{x}_n, y_n)\}$ with $U = \max_i [\mathbf{x}_i \cdot \mathbf{x}_i]$, Algorithm 1 terminates after incrementally adding at most $\max\left\{ \frac{2n}{\epsilon}, \frac{4nCU}{\epsilon^2} \right\}$ constraints to the constraint set $A^*$.*

The proof of Theorem 7 follows the procedures developed by Tsochantaridis et al. [28] for bounding the running time of structured support vector machines. We observe that this bound is quite loose in practice and the algorithm tends to converge much faster in our experiments.

# 4 Experiments

We evaluate the performance of the AL$^{0\text{-}1}$ classifier and compare with the three most popular multiclass SVM formulations: WW [11], CS [10], and LLW [12]. We use 12 datasets from the UCI Machine Learning repository [30] with various sizes and numbers of classes (details in Table 1). For each dataset, we consider the methods using the original feature space (linear kernel) and a kernelized feature space using the Gaussian radial basis function kernel.

Table 1: Properties of the datasets, the number of constraints considered by SVM models (WW/CS/LLW), the average number of constraints added to the constraint set for AL$^{0\text{-}1}$ and the average number of active constraints at the optima under both linear and Gausssian kernels.

| Dataset | Properties | | | | SVM constraints | AL$^{0\text{-}1}$ constraints added and active | | | |
|---|---|---|---|---|---|---|---|---|---|
| | # class | # train | # test | # feature | | Linear kernel | | Gauss. kernel | |
| (1) iris | 3 | 105 | 45 | 4 | 210 | 213 | 13 | 223 | 38 |
| (2) glass | 6 | 149 | 65 | 9 | 745 | 578 | 125 | 490 | 252 |
| (3) redwine | 10 | 1119 | 480 | 11 | 10071 | 5995 | 1681 | 3811 | 1783 |
| (4) ecoli | 8 | 235 | 101 | 7 | 1645 | 614 | 117 | 821 | 130 |
| (5) vehicle | 4 | 592 | 254 | 18 | 1776 | 1310 | 311 | 1201 | 248 |
| (6) segment | 7 | 1617 | 693 | 19 | 9702 | 4410 | 244 | 4312 | 469 |
| (7) sat | 7 | 4435 | 2000 | 36 | 26610 | 11721 | 1524 | 11860 | 6269 |
| (8) optdigits | 10 | 3823 | 1797 | 64 | 34407 | 7932 | 597 | 10072 | 2315 |
| (9) pageblocks | 5 | 3831 | 1642 | 10 | 15324 | 9459 | 427 | 9155 | 551 |
| (10) libras | 15 | 252 | 108 | 90 | 3528 | 1592 | 389 | 1165 | 353 |
| (11) vertebral | 3 | 217 | 93 | 6 | 434 | 344 | 78 | 342 | 86 |
| (12) breasttissue | 6 | 74 | 32 | 9 | 370 | 258 | 65 | 271 | 145 |

For our experimental methodology, we first make 20 random splits of each dataset into training and testing sets. We then perform two stage, five-fold cross validation on the training set of the first split to tune each model's parameter $C$ and the kernel parameter $\gamma$ under the kernelized formulation. In the first stage, the values for $C$ are $2^i, i = \{0, 3, 6, 9, 12\}$ and the values for $\gamma$ are $2^i, i = \{-12, -9, -6, -3, 0\}$. We select final values for $C$ from $2^i C_0, i = \{-2, -1, 0, 1, 2\}$ and values for $\gamma$ from $2^i \gamma_0, i = \{-2, -1, 0, 1, 2\}$ in the second stage, where $C_0$ and $\gamma_0$ are the best parameters obtained in the first stage. Using the selected parameters, we train each model on the 20 training sets and evaluate the performance on the corresponding testing set. We use the Shark machine learning library [31] for the implementation of the three multiclass SVM formulations.

Despite having an exponential number of possible constraints (i.e., $n(2^{|\mathcal{Y}|} - 1)$ for $n$ examples versus $n(|\mathcal{Y}| - 1)$ for SVMs), a much smaller number of constraints need to be considered by the AL$^{0\text{-}1}$ algorithm in practice to realize a better approximation ($\epsilon = 0$) than Theorem 7 provides. Table 1 shows how the total number of constraints for multiclass SVM compares to the number considered in practice by our AL$^{0\text{-}1}$ algorithm for linear and Gaussian kernel feature spaces. These range from a small fraction (0.23) of the SVM constraints for `optdigits` to a slightly greater number (with a fraction of 1.06) for `iris`. More specifically, of the over 3.9 million ($= 2^{10} \cdot 3823$) possible constraints for `optdigits` when training the classifier, fewer than $0.3\%$ (7932 or 10072 depending on the feature representation) are added to the constraint set during the constraint generation process. Fewer still (597 or 2315 constraints—less than $0.06\%$) are constraints that are active in the final classifier with non-zero dual parameters. The sparsity of the dual parameters provides a key computational benefit for support vector machines over logistic regression, which has essentially all non-zero dual parameters. The small number of active constraints shown in Table 1 demonstrate that AL$^{0\text{-}1}$ induces similar sparsity, providing efficiency when employed with kernel methods.

We report the accuracy of each method averaged over the 20 dataset splits for both linear feature representations and Gaussian kernel feature representations in Table 2. We denote the results that are either the best of all four methods or not worse than the best with statistical significance (under paired t-test with $\alpha = 0.05$) using bold font. We also show the accuracy averaged over all of the datasets for each method and the number of dataset for which each method is "indistinguishably best" (bold numbers) in the last row. As we can see from the table, the only alternative model that is Fisher

Table 2: The mean and (in parentheses) standard deviation of the accuracy for each model with linear kernel and Gaussian kernel feature representations. Bold numbers in each case indicate that the result is the best or not significantly worse than the best (paired t-test with $\alpha = 0.05$).

| D | Linear Kernel | | | | Gaussian Kernel | | | |
|---|---|---|---|---|---|---|---|---|
| | $AL^{0\text{-}1}$ | WW | CS | LLW | $AL^{0\text{-}1}$ | WW | CS | LLW |
| (1) | **96.3** (3.1) | **96.0** (2.6) | **96.3** (2.4) | 79.7 (5.5) | **96.7** (2.4) | **96.4** (2.4) | **96.2** (2.3) | 95.4 (2.1) |
| (2) | **62.5** (6.0) | **62.2** (3.6) | **62.5** (3.9) | 52.8 (4.6) | **69.5** (4.2) | 66.8 (4.3) | **69.4** (4.8) | **69.2** (4.4) |
| (3) | **58.8** (2.0) | **59.1** (1.9) | 56.6 (2.0) | 57.7 (1.7) | 63.3 (1.8) | 64.2 (2.0) | 64.2 (1.9) | **64.7** (2.1) |
| (4) | **86.2** (2.2) | 85.7 (2.5) | **85.8** (2.3) | 74.1 (3.3) | **86.0** (2.7) | 84.9 (2.4) | **85.6** (2.4) | **86.0** (2.5) |
| (5) | **78.8** (2.2) | **78.8** (1.7) | **78.4** (2.3) | 69.8 (3.7) | **84.3** (2.5) | **84.4** (2.6) | 83.8 (2.3) | **84.4** (2.6) |
| (6) | 94.9 (0.7) | 94.9 (0.8) | **95.2** (0.8) | 75.8 (1.5) | 96.5 (0.6) | **96.6** (0.5) | 96.3 (0.6) | 96.4 (0.5) |
| (7) | 84.9 (0.7) | **85.4** (0.7) | 84.7 (0.7) | 74.9 (0.9) | 91.9 (0.5) | **92.0** (0.6) | 91.9 (0.5) | **91.9** (0.4) |
| (8) | **96.6** (0.6) | 96.5 (0.7) | 96.3 (0.6) | 76.2 (2.2) | 98.7 (0.4) | 98.8 (0.4) | 98.8 (0.3) | **98.9** (0.3) |
| (9) | 96.0 (0.5) | 96.1 (0.5) | **96.3** (0.5) | 92.5 (0.8) | **96.8** (0.5) | 96.6 (0.4) | 96.7 (0.4) | 96.6 (0.4) |
| (10) | **74.1** (3.3) | 72.0 (3.8) | 71.3 (4.3) | 34.0 (6.4) | 83.6 (3.8) | 83.8 (3.4) | **85.0** (3.9) | 83.2 (4.2) |
| (11) | **85.5** (2.9) | **85.9** (2.7) | **85.4** (3.3) | 79.8 (5.6) | **86.0** (3.1) | **85.3** (2.9) | 85.5 (3.3) | 84.4 (2.7) |
| (12) | **64.4** (7.1) | 59.7 (7.8) | **66.3** (6.9) | 58.3 (8.1) | **68.4** (8.6) | **68.1** (6.5) | **66.6** (8.9) | **68.0** (7.2) |
| avg | 81.59 | 81.02 | 81.25 | 68.80 | 85.14 | 84.82 | 85.00 | 84.93 |
| #bold | 9 | 6 | 8 | 0 | 9 | 6 | 6 | 7 |

consistent—the LLW model—performs poorly on all datasets when only linear features are employed. This matches with previous experimental results conducted by Doğan et al. [15] and demonstrates a weakness of using an absolute margin for the loss function (rather than the relative margins of all other methods). The $AL^{0\text{-}1}$ classifier performs competitively with the WW and CS models with a slight advantages on overall average accuracy and a larger number of "indistinguishably best" performances on datasets—or, equivalently, fewer statistically significant losses to any other method.

The kernel trick in the Gaussian kernel case provides access to much richer feature spaces, improving the performance of all models, and the LLW model especially. In general, all models provide competitive results in the Gaussian kernel case. The $AL^{0\text{-}1}$ classifier maintains a similarly slight advantage and only provides performance that is sub-optimal (with statistical significance) in three of the twelve datasets versus six of twelve and five of twelve for the other methods. We conclude that the multiclass adversarial method performs well in both low and high dimensional feature spaces. Recalling the theoretical analysis of the adversarial method, it is a well-motivated (from the adversarial zero-one loss minimization) multiclass classifier that enjoys both strong theoretical properties (Fisher consistency and universal consistency) and empirical performance.

## 5   Conclusion

Generalizing support vector machines to multiclass settings in a theoretically sound manner remains a long-standing open problem. Though the loss function requirements guaranteeing Fisher-consistency are well-understood [13], the few Fisher-consistent classifiers that have been developed (e.g., LLW) often are not competitive with inconsistent multiclass classifiers in practice. In this paper, we have sought to fill this gap between theory and practice. We have demonstrated that multiclass adversarial classification under zero-one loss can be recast from an empirical risk minimization perspective and its surrogate loss, $AL^{0\text{-}1}$, shown to satisfy the Fisher consistency property, leading to a universally consistent classifier that also performs well in practice. We believe that this is an important contribution in understanding both adversarial methods and the generalized hinge loss. Our future work includes investigating the adversarial methods under the different losses and exploring other theoretical properties of the adversarial framework, including generalization bounds.

## Acknowledgments

This research was supported as part of the Future of Life Institute (futureoflife.org) FLI-RFP-AI1 program, grant#2016-158710 and by NSF grant RI-#1526379.

## Footnotes

[1]Farnia & Tse independently and concurrently discovered this same loss function [20]. They provide an analysis focused on generalization bounds and experiments for binary classification.

[2]The proof of this theorem and others in the paper are contained in the Supplementary Materials.

[3]We refer the reader to Appendix H for a comparison of the binary adversarial method and the binary SVM.

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
