[Supplementary Material]

# Supplementary Materials

## A   Proof for the Adversarial Zero-One Loss (Theorem 1)

To prove Theorem 1, which defines the ERM loss of the adversarial classifier for zero-one loss, we first develop two important lemmas. Our approach analyzes the Nash equilibrium value of the game described in Eq. 3, beginning with a specific simple case of the game: the game matrix $\mathbf{L}'_{\mathbf{x}_i,\theta}$ is a completely mixed game.

**Lemma 1.** *If the game matrix $\mathbf{L}'_{\mathbf{x}_i,\theta}$ is a completely mixed game, i.e., every adversary's and predictor's strategy has non zero probability, and if the game value is nonzero, then the equilibrium game value for the game is $\frac{\sum_{j=1}^{|\mathcal{Y}|} \psi_{j,y_i}(\mathbf{x}_i)+|\mathcal{Y}|-1}{|\mathcal{Y}|}$.*

*Proof.* According to Barron [3], a completely mixed game with a square game matrix has only one saddle point. If we know that the game value is nonzero, then the game matrix is invertible and the game value can be computed using the formula $v(M) = \frac{1}{J^T M^{-1} J}$, where $M$ is the zero-sum game matrix with row player as the maximizing player and column player as the minimizing player, and $J$ is a vector with length $|\mathcal{Y}|$ containing all ones, $J = [1, 1, \ldots, 1]^T$.

Therefore, under the adversarial game matrix formulation, $M$ is the transpose of $\mathbf{L}_{\mathbf{x}_i,\theta}$:

$$M = \mathbf{L}'^{\mathrm{T}}_{\mathbf{x}_i,\theta} = \begin{bmatrix} \psi_{1,y_i}(\mathbf{x}_i) & \psi_{1,y_i}(\mathbf{x}_i)+1 & \cdots & \psi_{1,y_i}(\mathbf{x}_i)+1 \\ \psi_{2,y_i}(\mathbf{x}_i)+1 & \psi_{2,y_i}(\mathbf{x}_i) & \cdots & \psi_{2,y_i}(\mathbf{x}_i)+1 \\ \vdots & \vdots & \ddots & \vdots \\ \psi_{|\mathcal{Y}|,y_i}(\mathbf{x}_i)+1 & \psi_{|\mathcal{Y}|,y_i}(\mathbf{x}_i)+1 & \cdots & \psi_{|\mathcal{Y}|,y_i}(\mathbf{x}_i) \end{bmatrix}. \tag{9}$$

The inverse of game matrix $M$ is of the form (detailed proof described in Appendix G):

$$M^{-1} = \begin{bmatrix} a_{1,1} & a_{1,2} & \cdots & a_{1,|\mathcal{Y}|} \\ a_{2,1} & a_{2,2} & \cdots & a_{2,|\mathcal{Y}|} \\ \vdots & \vdots & \ddots & \vdots \\ a_{|\mathcal{Y}|,1} & a_{|\mathcal{Y}|,2} & \cdots & a_{|\mathcal{Y}|,|\mathcal{Y}|} \end{bmatrix}, \tag{10}$$

where:

$$a_{k,k} = -\frac{\sum_{j=1,j\neq k}^{|\mathcal{Y}|} \psi_{j,y_i}(\mathbf{x}_i) + |\mathcal{Y}| - 2}{\sum_{j=1}^{|\mathcal{Y}|} \psi_{j,y_i}(\mathbf{x}_i) + |\mathcal{Y}| - 1} \quad k \in \{1, \ldots, |\mathcal{Y}|\},$$

$$a_{k,l} = \frac{\psi_{k,y_i}(\mathbf{x}_i) + 1}{\sum_{j=1}^{|\mathcal{Y}|} \psi_{j,y_i}(\mathbf{x}_i) + |\mathcal{Y}| - 1} \quad k,l \in \{1, \ldots, |\mathcal{Y}|\}, k \neq l.$$

The value of the vector-matrix multiplication $J^T M^{-1} J$ where $J$ is a vector containing all ones, is the summation of all elements in $M^{-1}$:

$$J^T M^{-1} J = \frac{-\sum_{k=1}^{|\mathcal{Y}|} \left[ \sum_{\substack{j=1 \\ j\neq k}}^{|\mathcal{Y}|} \psi_{j,y_i}(\mathbf{x}_i) + |\mathcal{Y}| - 2 \right] + \sum_{\substack{k,l\in\{1,\ldots,|\mathcal{Y}|\} \\ k\neq l}} [\psi_{k,y_i}(\mathbf{x}_i) + 1]}{\sum_{j=1}^{|\mathcal{Y}|} \psi_{j,y_i}(\mathbf{x}_i) + |\mathcal{Y}| - 1} \tag{11}$$

$$= \frac{\sum_{\substack{k,l\in\{1,\ldots,|\mathcal{Y}|\} \\ k\neq l}} \psi_{k,y_i}(\mathbf{x}_i) - \sum_{\substack{j,k\in\{1,\ldots,|\mathcal{Y}|\} \\ j\neq k}} \psi_{j,y_i}(\mathbf{x}_i) + |\mathcal{Y}|(|\mathcal{Y}|-1) - |\mathcal{Y}|(|\mathcal{Y}|-2)}{\sum_{j=1}^{|\mathcal{Y}|} \psi_{j,y_i}(\mathbf{x}_i) + |\mathcal{Y}| - 1} \tag{12}$$

$$= \frac{|\mathcal{Y}|}{\sum_{j=1}^{|\mathcal{Y}|} \psi_{j,y_i}(\mathbf{x}_i) + |\mathcal{Y}| - 1}. \tag{13}$$

Therefore, the equilibrium game value when the game matrix $\mathbf{L}'_{\mathbf{x}_i,\theta}$ is a completely mixed game with nonzero game value is:

$$v(M) = \frac{1}{\frac{|\mathcal{Y}|}{\sum_{j=1}^{|\mathcal{Y}|} \psi_{j,y_i}(\mathbf{x}_i)+|\mathcal{Y}|-1}} = \frac{\sum_{j=1}^{|\mathcal{Y}|} \psi_{j,y_i}(\mathbf{x}_i)+|\mathcal{Y}|-1}{|\mathcal{Y}|}. \tag{14}$$

$\square$

We next consider the case where one of the adversary's strategies in the game matrix $\mathbf{L}'_{\mathbf{x}_i,\theta}$ has zero probability. Lemma 2 establishes the game value for such cases.

**Lemma 2.** *If an adversary strategy $k$ (corresponding with column $k$) has zero probability in the Nash equilibrium of game matrix $\mathbf{L}'_{\mathbf{x}_i,\theta}$, and if the game matrix excluding column and row $k$ is a completely mixed game with nonzero game value, the equilibrium value for the game is $\frac{\sum_{j=1,j\neq k}^{|\mathcal{Y}|} \psi_{j,y_i}(\mathbf{x}_i)+|\mathcal{Y}|-2}{|\mathcal{Y}|-1} \geq \frac{\sum_{j=1}^{|\mathcal{Y}|} \psi_{j,y_i}(\mathbf{x}_i)+|\mathcal{Y}|-1}{|\mathcal{Y}|}.$*

*Proof.* If an adversary strategy $k$ (corresponds with column $k$) has zero probability, row $k$ also has zero probability because removing column $k$ causes the removal of $\psi_{k,y_i}(\mathbf{x}_i)$ term from row $k$, leaving $\psi_{j,y_i}(\mathbf{x}_i) + 1, \forall j \neq k$. Since the row player seeks to minimize the game value and row $k$ now has values greater than or equal to the other rows, row $k$ can be dominated and therefore has zero probability in the Nash equilibrium of the game.

Using Lemma 1, we can compute the value of a game without column $k$ and row $k$ in the case that the game value is nonzero, which is $\frac{\sum_{j=1,j\neq k}^{|\mathcal{Y}|} \psi_{j,y_i}(\mathbf{x}_i)+|\mathcal{Y}|-2}{|\mathcal{Y}|-1}$. Therefore, since both column and row $k$ has zero probability and the game matrix excluding column and row $k$ is a completely mixed game, the value of the game matrix $\mathbf{L}'_{\mathbf{x}_i,\theta}$ is also $\frac{\sum_{j=1,j\neq k}^{|\mathcal{Y}|} \psi_{j,y_i}(\mathbf{x}_i)+|\mathcal{Y}|-2}{|\mathcal{Y}|-1}$. By definition of equilibrium game value, this value has to be greater than or equal to any possible strategy that the adversary player can play, including the strategy that assigns non-zero probability to column and row $k$. Therefore, we can also conclude that the inequality $\frac{\sum_{j=1,j\neq k}^{|\mathcal{Y}|} \psi_{j,y_i}(\mathbf{x}_i)+|\mathcal{Y}|-2}{|\mathcal{Y}|-1} \geq \frac{\sum_{j=1}^{|\mathcal{Y}|} \psi_{j,y_i}(\mathbf{x}_i)+|\mathcal{Y}|-1}{|\mathcal{Y}|}$ holds. $\square$

The proof for Theorem 1 involves the generalization of Lemma 2 to all possible combination of strategies with zero probability, as described in the following.

**Theorem 1.** *The model parameters $\theta$ for multiclass zero-one adversarial classification are equivalently obtained from empirical risk minimization under the adversarial zero-one loss function:*

$$AL_{\mathbf{f}}^{0\text{-}1}(\mathbf{x}_i, y_i) = \max_{\substack{\mathcal{S}\subseteq\{1,\dots,|\mathcal{Y}|\} \\ \mathcal{S}\neq\emptyset}} \frac{\sum_{j\in\mathcal{S}} \psi_{j,y_i}(\mathbf{x}_i)+|\mathcal{S}|-1}{|\mathcal{S}|}, \tag{15}$$

*where $\mathcal{S}$ is any non-empty member of the powerset of classes $\{1, 2, \dots, |\mathcal{Y}|\}$.*

*Proof.* Generalizing Lemma 2, if we consider all possible combination of strategies with zero probability, then the game value of game matrix $\mathbf{L}'_{\mathbf{x}_i,\theta}$ equals to the game value of the matrix after removing all columns and rows with zero probability, resulting in a completely mixed game matrix. Let $R$ be a set of the remaining columns in the resulting completely mixed game matrix. If we know that the game value for the resulting game matrix is nonzero, then the equilibrium game value of $\mathbf{L}'_{\mathbf{x}_i,\theta}$ is $\frac{\sum_{j\in R} \psi_{j,y_i}(\mathbf{x}_i)+|R|-1}{|R|}$. This value must be greater than or equal to any possible strategy of the adversary player. Moreover, we know that if the set $R$ contains only one element $\{y_i\}$, then the game value is

$$\frac{\sum_{j\in R} \psi_{j,y_i}(\mathbf{x}_i)+|R|-1}{|R|} = \frac{\psi_{y_i,y_i}(\mathbf{x}_i)+1-1}{1} = \theta^T\left(\phi(\mathbf{x}_i,y_i)-\phi(\mathbf{x}_i,y_i)\right) = 0. \tag{16}$$

Therefore, by considering all possible combination of strategies with zero probability, we can conclude that the game value of the game matrix $\mathbf{L}'_{\mathbf{x}_i,\theta}$ is the following function:

$$\max_{\substack{\mathcal{S}\subseteq\{1,\dots,|\mathcal{Y}|\} \\ \mathcal{S}\neq\emptyset}} \frac{\sum_{j\in\mathcal{S}} \psi_{j,y_i}(\mathbf{x}_i)+|\mathcal{S}|-1}{|\mathcal{S}|}. \tag{17}$$

The adversarial optimization (Eq. 3) can be viewed from the empirical risk minimization perspective where the loss function is defined by the game value described above, hence proving the theorem. □

# B  Proof for the Consistency Analysis (Theorem 2 and Theorem 3)

In this section, we will prove Theorem 2 and Theorem 3. We first analyze the properties of the set of labels that define the supporting hyperplane of $AL^{0\text{-}1}$ loss.

**Lemma 3.** *If $R^* \subseteq \{1, \ldots, |\mathcal{Y}|\}$ is the set of labels that defines the supporting hyperplane of $AL^{0\text{-}1}$ loss (Eq. (4)) when the true label $y = k$, then $R^*$ also defines the supporting hyperplane of $AL^{0\text{-}1}$ loss when the true label is any label other than $k$.*

*Proof.* We know that for any set $R \subseteq \{1, \ldots, |\mathcal{Y}|\}$,

$$\frac{\sum_{j \in R} \psi_{j,y}(\mathbf{x}) + |R| - 1}{|R|} = \frac{\sum_{j \in R} [f_j(\mathbf{x}) - f_y(\mathbf{x})] + |R| - 1}{|R|} \tag{18}$$

$$= \frac{\sum_{j \in R} f_j(\mathbf{x}) + |R| - 1}{|R|} - f_y(\mathbf{x}). \tag{19}$$

Since $R^*$ is the set of labels that define the supporting hyperplane of the loss when the true label class $y = k$, then for any other set $R \subseteq \{1, \ldots, |\mathcal{Y}|\}$:

$$\frac{\sum_{j \in R^*} \psi_{j,k}(\mathbf{x}) + |R^*| - 1}{|R^*|} \geq \frac{\sum_{j \in R} \psi_{j,k}(\mathbf{x}) + |R| - 1}{|R|} \tag{20}$$

$$\frac{\sum_{j \in R^*} f_j(\mathbf{x}) + |R^*| - 1}{|R^*|} - f_k(\mathbf{x}) \geq \frac{\sum_{j \in R} f_j(\mathbf{x}) + |R| - 1}{|R|} - f_k(\mathbf{x}) \tag{21}$$

$$\frac{\sum_{j \in R^*} f_j(\mathbf{x}) + |R^*| - 1}{|R^*|} \geq \frac{\sum_{j \in R} f_j(\mathbf{x}) + |R| - 1}{|R|}. \tag{22}$$

Therefore, for any other class $l$, the inequality below also holds:

$$\frac{\sum_{j \in R^*} f_j(\mathbf{x}) + |R^*| - 1}{|R^*|} - f_l(\mathbf{x}) \geq \frac{\sum_{j \in R} f_j(\mathbf{x}) + |R| - 1}{|R|} - f_l(\mathbf{x}) \tag{23}$$

$$\frac{\sum_{j \in R^*} \psi_{j,l}(\mathbf{x}) + |R^*| - 1}{|R^*|} \geq \frac{\sum_{j \in R} \psi_{j,l}(\mathbf{x}) + |R| - 1}{|R|}. \tag{24}$$

□

We now analyze $AL^{0\text{-}1}$ using a geometrical view. We know that the loss is the maximization over different linear hyperplanes. We analyze the hyperplane defined by the complete set of labels $R^* = \{1, \ldots, |\mathcal{Y}|\}$. For three class classification (Figure 2a), it is the hyperplane in the middle with $AL^{0\text{-}1}$ value $\frac{\psi_{1,y}(\mathbf{x}) + \psi_{2,y}(\mathbf{x}) + \psi_{3,y}(\mathbf{x}) + 2}{3}$. Note that in Figure 2a, $\psi_{1,y}(\mathbf{x}) = 0$ since $y = 1$. We demonstrate the circumstances that correspond with this case in the following lemma.

**Lemma 4.** *The hyperplane defined by the complete set of labels $R^* = \{1, \ldots, |\mathcal{Y}|\}$ supports $AL^{0\text{-}1}$ in the area where $-\frac{1}{|\mathcal{Y}|} \leq f_j(\mathbf{x}) \leq \frac{|\mathcal{Y}| - 1}{|\mathcal{Y}|}, \forall j \in \{1, \ldots, |\mathcal{Y}|\}$ given that $\sum_{j=1}^{|\mathcal{Y}|} f_j(\mathbf{x}) = 0$.*

*Proof.* Since the hyperplane defined by the complete set of labels $R^* = \{1, \ldots, |\mathcal{Y}|\}$ supports $AL^{0\text{-}1}$, from the proof in Lemma 3, we know that:

$$\frac{\sum_{j=1}^{|\mathcal{Y}|} f_j(\mathbf{x}) + |\mathcal{Y}| - 1}{|\mathcal{Y}|} = \frac{|\mathcal{Y}| - 1}{|\mathcal{Y}|} \geq \frac{\sum_{j \in R} f_j(\mathbf{x}) + |R| - 1}{|R|}, \tag{25}$$

for any $R \subseteq \{1, \ldots, |\mathcal{Y}|\}, R \neq \emptyset$. In the case that $R$ contains only one element $j$, we know that $\frac{|\mathcal{Y}| - 1}{|\mathcal{Y}|} \geq f_j(\mathbf{x})$. In the case that $R$ contains all element but $j$, we have:

$$\frac{|\mathcal{Y}| - 1}{|\mathcal{Y}|} \geq \frac{\sum_{k \in \{1, \ldots, |\mathcal{Y}|\}, k \neq j} f_k(\mathbf{x}) + |\mathcal{Y}| - 2}{|\mathcal{Y}| - 1} = \frac{-f_j(\mathbf{x}) + |\mathcal{Y}| - 2}{|\mathcal{Y}| - 1} \tag{26}$$

$$|\mathcal{Y}|^2 - 2|\mathcal{Y}| + 1 \geq -|\mathcal{Y}|f_j(\mathbf{x}) + |\mathcal{Y}|^2 - 2|\mathcal{Y}| \tag{27}$$

$$f_j(\mathbf{x}) \geq -\frac{1}{|\mathcal{Y}|}. \tag{28}$$

In general for any set $R \subseteq \{1, \ldots, |\mathcal{Y}|\}, R \neq \emptyset$, the following holds:

$$\frac{|\mathcal{Y}| - 1}{|\mathcal{Y}|} \geq \frac{\sum_{j \in R} f_j(\mathbf{x}) + |R| - 1}{|R|} \tag{29}$$

$$|R||\mathcal{Y}| - |R| \geq |\mathcal{Y}| \sum_{j \in R} f_j(\mathbf{x}) + |R||\mathcal{Y}| - |\mathcal{Y}| \tag{30}$$

$$\sum_{j \in R} f_j(\mathbf{x}) \leq \frac{|\mathcal{Y}| - |R|}{|\mathcal{Y}|}. \tag{31}$$

If we consider $R^c$ as the complement of set $R$, i.e., $R^c = \{1, \ldots, |\mathcal{Y}|\} \backslash R$, the following also holds:

$$\frac{|\mathcal{Y}| - 1}{|\mathcal{Y}|} \geq \frac{\sum_{j \in R^c} f_j(\mathbf{x}) + |R^c| - 1}{|R^c|} = \frac{-\sum_{j \in R} f_j(\mathbf{x}) + |R^c| - 1}{|R^c|} \tag{32}$$

$$|R^c||\mathcal{Y}| - |R^c| \geq -|\mathcal{Y}| \sum_{j \in R} f_j(\mathbf{x}) + |R^c||\mathcal{Y}| - |\mathcal{Y}| \tag{33}$$

$$\sum_{j \in R} f_j(\mathbf{x}) \geq -\frac{|\mathcal{Y}| - |R^c|}{|\mathcal{Y}|} = -\frac{|R|}{|\mathcal{Y}|}. \tag{34}$$

We can easily see that the general case above is automatically implied from the individual rule $-\frac{1}{|\mathcal{Y}|} \leq f_j(\mathbf{x}) \leq \frac{|\mathcal{Y}|-1}{|\mathcal{Y}|}, \forall j \in \{1, \ldots, |\mathcal{Y}|\}$, given that $\sum_{j=1}^{|\mathcal{Y}|} f_j(\mathbf{x}) = 0$. $\qquad \square$

Next, we prove Theorem 2, which states that the loss for minimizer $\mathbf{f}^*$ of $\mathbb{E}\left[\mathrm{AL}_{\mathbf{f}}^{0\text{-}1}(\mathbf{X}, Y)|\mathbf{X} = \mathbf{x}\right]$ resides in the area described in Lemma 4.

**Theorem 2.** *The loss for the minimizer $\mathbf{f}^*$ of $\mathbb{E}\left[AL_{\mathbf{f}}^{0\text{-}1}(\mathbf{X}, Y)|\mathbf{X} = \mathbf{x}\right]$ resides on the hyperplane defined (in Eq. 4) by the complete set of labels, $\mathcal{S} = \{1, \ldots, |\mathcal{Y}|\}$.*

*Proof.* We start the proof by denoting $R$ as a non-complete set of labels, $R \subsetneq \{1, \ldots, |\mathcal{Y}|\}, R \neq \emptyset$, that defines the supporting hyperplane of $\mathrm{AL}^{0\text{-}1}$ loss. Let $\mathbf{f}^0$ be the potential function where its loss resides on the hyperplane defined by $R$. We will show that we can construct $\mathbf{f}^1$ such that its loss resides on the hyperplane defined by the complete set of labels, $\mathcal{S} = \{1, \ldots, |\mathcal{Y}|\}$, such that $\mathbb{E}\left[\mathrm{AL}_{\mathbf{f}^1}^{0\text{-}1}(\mathbf{X}, Y)|\mathbf{X} = \mathbf{x}\right] \leq \mathbb{E}\left[\mathrm{AL}_{\mathbf{f}^0}^{0\text{-}1}(\mathbf{X}, Y)|\mathbf{X} = \mathbf{x}\right]$. Note that $\mathbb{E}\left[\mathrm{AL}_{\mathbf{f}}^{0\text{-}1}(\mathbf{X}, Y)|\mathbf{X} = \mathbf{x}\right] = \sum_{y=1}^{|\mathcal{Y}|} P_y(\mathbf{x}) \mathrm{AL}_{\mathbf{f}}^{0\text{-}1}(\mathbf{x}, y)$. In this proof, we need consider the loss for each possible true label $y \in \mathcal{Y}$.

Let $R^c$ be the complement of the set $R$, i.e., $R^c = \{1, \ldots, |\mathcal{Y}|\} \backslash R$, and let us denote $\psi_{j,y}^{\mathbf{f}^0}(\mathbf{x}) = f_j^0(\mathbf{x}) - f_y^0(\mathbf{x})$. We note that for $y \in R$, the loss, $\frac{\sum_{j \in R} \psi_{j,y}^{\mathbf{f}^0}(\mathbf{x}) + |R| - 1}{|R|}$, does not depend on any $\psi_{k,y}^{\mathbf{f}^0}(\mathbf{x})$ for $k \in R^c$. Therefore, changing any $\psi_{k,y}^{\mathbf{f}^0}(\mathbf{x})$ where $k \in R^c$ does not change the loss when the true label is $y \in R$.

Let $\mathbf{f}^1$ be the potential function such that $f_k^1(\mathbf{x}) = -\frac{1}{|\mathcal{Y}|}$ for $\forall k \in R^c$, and keep all $\psi_{j,y}^{\mathbf{f}^1}(\mathbf{x}) = f_j^1(\mathbf{x}) - f_y^1(\mathbf{x})$ remaining the same as $\psi_{j,y}^{\mathbf{f}^0}(\mathbf{x}) = f_j^0(\mathbf{x}) - f_y^0(\mathbf{x})$ for $j \in R$ and $y \in R$. Let $b = -\frac{|R^c|}{|\mathcal{Y}|} - \sum_{k \in R^c} f_k^0(\mathbf{x})$, setting $f_j^1(\mathbf{x}) = f_j^0(\mathbf{x}) - \frac{b}{|R|}$ for all $j \in R$ will satisfy the requirement above while keeping it valid, i.e., $\sum_{j=1}^{|\mathcal{Y}|} f_j^1(\mathbf{x}) = 0$. Analyzing the transformation above, we have:

$$\frac{\sum_{j \in R} f_j^1(\mathbf{x}) + |R| - 1}{|R|} = \frac{\sum_{j \in R} f_j^0(\mathbf{x}) + \sum_{j \in R^c} f_j^0(\mathbf{x}) + \frac{|R^c|}{|\mathcal{Y}|} + |R| - 1}{|R|} \tag{35}$$

$$= \frac{|R^c| + |R||\mathcal{Y}| - |\mathcal{Y}|}{|R||\mathcal{Y}|} = \frac{|R||\mathcal{Y}| - |R|}{|R||\mathcal{Y}|} = \frac{|\mathcal{Y}| - 1}{|\mathcal{Y}|} \tag{36}$$

$$= \frac{\sum_{j=1}^{|\mathcal{Y}|} f_j^1(\mathbf{x}) + |\mathcal{Y}| - 1}{|\mathcal{Y}|}. \tag{37}$$

We can view this transformation as the following: when $y \in R$, we fix $\psi_{j,y}(\mathbf{x})$ for all $j \in R$ and move all $\psi_{k,y}(\mathbf{x})$ for all $k \in R^c$ towards the intersection between the hyperplane defined by $R$ and the hyperplane defined by the complete set of labels.

We know that $\text{AL}_{\mathbf{f}^1}^{0\text{-}1}(\mathbf{x}, y)$ is equal to $\text{AL}_{\mathbf{f}^0}^{0\text{-}1}(\mathbf{x}, y)$ if the true label $y \in R$. The difference comes when the true label $y \in R^c$. For $\mathbf{f}^1$, the loss will be:

$$\frac{\sum_{j \in R} f_j^1(\mathbf{x}) + |R| - 1}{|R|} - f_y^1(\mathbf{x}) = \frac{\mathcal{Y} - 1}{\mathcal{Y}} + \frac{1}{\mathcal{Y}} = 1. \tag{38}$$

Before analyzing the loss for $\mathbf{f}^0$, we observe the following inequality. Since $R$ is the set that defines the hyperplane that supports the $\text{AL}^{0\text{-}1}$ loss under $\mathbf{f}^0$, then for all $k \in R^c$:

$$\frac{\sum_{j \in R} f_j^0(\mathbf{x}) + |R| - 1}{|R|} \geq \frac{\sum_{j \in R} f_j^0(\mathbf{x}) + f_k^0(\mathbf{x}) + |R|}{|R| + 1} \tag{39}$$

$$|R| \sum_{j \in R} f_j^0(\mathbf{x}) + |R|^2 - R + \sum_{j \in R} f_j^0(\mathbf{x}) + |R| - 1 \geq |R| \sum_{j \in R} f_j^0(\mathbf{x}) + |R| f_k^0(\mathbf{x}) + |R|^2 \tag{40}$$

$$\sum_{j \in R} f_j^0(\mathbf{x}) - 1 \geq |R| f_k^0(\mathbf{x}) \tag{41}$$

$$\sum_{j \in R} f_j^0(\mathbf{x}) - |R| f_k^0(\mathbf{x}) \geq 1. \tag{42}$$

Applying the inequality above, we get the loss for $\mathbf{f}^0$ when the true label $y \in R^c$:

$$\frac{\sum_{j \in R} f_j^0(\mathbf{x}) + |R| - 1}{|R|} - f_y^0(\mathbf{x}) = \frac{\sum_{j \in R} f_j^0(\mathbf{x}) - |R| f_y^0(\mathbf{x}) + |R| - 1}{|R|} \tag{43}$$

$$\geq \frac{1 + |R| - 1}{|R|} = 1. \tag{44}$$

In the analysis above, we construct $\mathbf{f}^1$, where its loss resides in the intersection between the hyperplane defined by $R$ and the hyperplane defined by the complete set of labels. Since $\mathbb{E}\left[\text{AL}_{\mathbf{f}}^{0\text{-}1}(\mathbf{X}, Y)|\mathbf{X} = \mathbf{x}\right] = \sum_{y=1}^{|\mathcal{Y}|} P_y(\mathbf{x}) \text{AL}_{\mathbf{f}}^{0\text{-}1}(\mathbf{x}, y)$, the analysis above shows that $\mathbb{E}\left[\text{AL}_{\mathbf{f}^1}^{0\text{-}1}(\mathbf{X}, Y)|\mathbf{X} = \mathbf{x}\right] \leq \mathbb{E}\left[\text{AL}_{\mathbf{f}^0}^{0\text{-}1}(\mathbf{X}, Y)|\mathbf{X} = \mathbf{x}\right]$. Therefore, we can conclude that given the probability for each class $P_y(\mathbf{x})$, the loss of the minimizer of $\mathbb{E}\left[\text{AL}_{\mathbf{f}}^{0\text{-}1}(\mathbf{X}, Y)|\mathbf{X} = \mathbf{x}\right]$ resides on the hyperplane defined by the complete set of labels, $\mathcal{S} = \{1, \ldots, |\mathcal{Y}|\}$. $\qquad\square$

To better understand Theorem 2, we will discuss an example for three-class classification. Let the potential function $\mathbf{f}^0 = [\frac{1}{6}, \frac{4}{6}, -\frac{5}{6}]$, whose loss resides on the hyperplane defined by the set of labels $R = \{1, 2\}$, i.e., $\frac{\psi_{1,y}(\mathbf{x}) + \psi_{2,y}(\mathbf{x}) + 1}{2}$. Figure 3 shows the plot of the loss when the true label $y$ is 1, 2 or 3. We can compute the losses as follows:

|  | $y = 1$ | $y = 2$ | $y = 3$ |
|---|---|---|---|
| $f_1^0(\mathbf{x}) = \frac{1}{6}$ | $\psi_{1,1}^{\mathbf{f}^0} = 0$ | $\psi_{1,2}^{\mathbf{f}^0} = -0.5$ | $\psi_{1,3}^{\mathbf{f}^0} = 1$ |
| $f_2^0(\mathbf{x}) = \frac{4}{6}$ | $\psi_{2,1}^{\mathbf{f}^0} = 0.5$ | $\psi_{2,2}^{\mathbf{f}^0} = 0$ | $\psi_{2,3}^{\mathbf{f}^0} = 1.5$ |
| $f_3^0(\mathbf{x}) = -\frac{5}{6}$ | $\psi_{3,1}^{\mathbf{f}^0} = -1$ | $\psi_{3,2}^{\mathbf{f}^0} = -1.5$ | $\psi_{3,3}^{\mathbf{f}^0} = 0$ |
| loss | $\text{AL}_{\mathbf{f}^0}^{0\text{-}1}(\mathbf{x}, 1) = 0.75$ | $\text{AL}_{\mathbf{f}^0}^{0\text{-}1}(\mathbf{x}, 2) = 0.25$ | $\text{AL}_{\mathbf{f}^0}^{0\text{-}1}(\mathbf{x}, 3) = 1.75$ |

We construct $\mathbf{f}^1$ from $\mathbf{f}^0$ using the steps described above. Note that $R^c = \{3\}$ and $b = -\frac{|R^c|}{|\mathcal{Y}|} - \sum_{k \in R^c} f_k^0(\mathbf{x}) = -\frac{1}{3} + \frac{5}{6} = \frac{1}{2}$. We set $f_3^1(\mathbf{x}) = -\frac{1}{|\mathcal{Y}|} = -\frac{1}{3}$ since $3 \in R^c$. We compute $f_1^1(\mathbf{x})$ and $f_2^1(\mathbf{x})$ by subtracting $f_1^0(\mathbf{x})$ and $f_2^0(\mathbf{x})$ with $\frac{b}{|R|} = \frac{1}{2 \cdot 2} = \frac{1}{4}$. The losses incurred by $\mathbf{f}^1$ (Figure 4) and its computations displayed in the following:

| | $y = 1$ | $y = 2$ | $y = 3$ |
|---|---|---|---|
| $f_1^1(\mathbf{x}) = \frac{1}{6} - \frac{1}{4} = -\frac{1}{12}$ | $\psi_{1,1}^{\mathbf{f}^1} = 0$ | $\psi_{1,2}^{\mathbf{f}^1} = -0.5$ | $\psi_{1,3}^{\mathbf{f}^1} = 0.25$ |
| $f_2^1(\mathbf{x}) = \frac{4}{6} - \frac{1}{4} = \frac{5}{12}$ | $\psi_{2,1}^{\mathbf{f}^1} = 0.5$ | $\psi_{2,2}^{\mathbf{f}^1} = 0$ | $\psi_{2,3}^{\mathbf{f}^1} = 0.75$ |
| $f_3^1(\mathbf{x}) = -\frac{1}{3}$ | $\psi_{3,1}^{\mathbf{f}^1} = -0.25$ | $\psi_{3,2}^{\mathbf{f}^1} = -0.75$ | $\psi_{3,3}^{\mathbf{f}^1} = 0$ |
| loss | $\mathrm{AL}_{\mathbf{f}^1}^{\text{0-1}}(\mathbf{x}, 1) = 0.75$ | $\mathrm{AL}_{\mathbf{f}^1}^{\text{0-1}}(\mathbf{x}, 2) = 0.25$ | $\mathrm{AL}_{\mathbf{f}^1}^{\text{0-1}}(\mathbf{x}, 3) = 1$ |

Figure 3: The plot of loss $\mathrm{AL}^{\text{0-1}}$ for $\mathbf{f}^0$ when the true label $y$ is 1, 2, or 3.

Figure 4: The plot of loss $\mathrm{AL}^{\text{0-1}}$ for $\mathbf{f}^1$ when the true label $y$ is 1, 2, or 3.

As we can see from the table, the loss incurred by $\mathbf{f}^1$ when the true label is 1 or 2 remains the same as the loss incurred by $\mathbf{f}^0$. The difference comes when the true label is 3. In this case, $\mathbf{f}^1$ incurs less loss than $\mathbf{f}^0$ (1 compared to 1.75).

Utilizing the lemmas and theorem above, we can now prove the Fisher consistency of $\mathrm{AL}^{\text{0-1}}$.

**Theorem 3.** *The adversarial zero-one loss, $\mathrm{AL}^{\text{0-1}}$, from Eq.* (4) *is Fisher consistent.*

*Proof.* For any given $\mathbf{X} = \mathbf{x}$, our goal is to minimize $\mathbb{E}\left[\mathrm{AL}_{\mathbf{f}}^{\text{0-1}}(\mathbf{X}, Y)|\mathbf{X} = \mathbf{x}\right] = \sum_{y=1}^{|\mathcal{Y}|} P_y(\mathbf{x}) \max_{\substack{\mathcal{S} \subseteq \{1, \ldots, |\mathcal{Y}|\} \\ \mathcal{S} \neq \emptyset}} \frac{\sum_{j \in \mathcal{S}} \psi_{j,y}(\mathbf{x}) + |\mathcal{S}| - 1}{|\mathcal{S}|}$. According to Theorem 2, it is equal to minimizing the following:

$$\sum_{y=1}^{|\mathcal{Y}|} P_y(\mathbf{x}) \frac{\sum_{j=1}^{|\mathcal{Y}|} \psi_{j,y}(\mathbf{x}) + |\mathcal{Y}| - 1}{|\mathcal{Y}|} = \sum_{y=1}^{|\mathcal{Y}|} P_y(\mathbf{x}) \left[ \frac{\sum_{j=1}^{|\mathcal{Y}|} (f_j(\mathbf{x}) - f_y(\mathbf{x})) + |\mathcal{Y}| - 1}{|\mathcal{Y}|} \right] \tag{45}$$

$$= \sum_{y=1}^{|\mathcal{Y}|} P_y(\mathbf{x}) \left[ \frac{|\mathcal{Y}| - 1}{|\mathcal{Y}|} - f_y(\mathbf{x}) \right] \tag{46}$$

$$= \frac{|\mathcal{Y}| - 1}{|\mathcal{Y}|} \sum_{y=1}^{|\mathcal{Y}|} P_y(\mathbf{x}) - \sum_{y=1}^{|\mathcal{Y}|} P_y(\mathbf{x}) f_y(\mathbf{x}) \tag{47}$$

$$= \frac{|\mathcal{Y}| - 1}{|\mathcal{Y}|} - \sum_{y=1}^{|\mathcal{Y}|} P_y(\mathbf{x}) f_y(\mathbf{x}), \tag{48}$$

subject to $-\frac{1}{|\mathcal{Y}|} \leq f_j(\mathbf{x}) \leq \frac{|\mathcal{Y}|-1}{|\mathcal{Y}|}$ and $\sum_{j=1}^{|\mathcal{Y}|} f_j(\mathbf{x}) = 0$.

Since $(|\mathcal{Y}|-1)/|\mathcal{Y}|$ is constant with respect to $\{f_i\}$, finding $\mathbf{f}^*$ in the minimization above is equivalent with finding $\mathbf{f}^*$ in the following maximization:

$$\max_{\mathbf{f}} \quad \sum_{y=1}^{|\mathcal{Y}|} P_y(\mathbf{x}) f_y(\mathbf{x}) \tag{49}$$

$$\text{subject to} \quad -\frac{1}{|\mathcal{Y}|} \leq f_j(\mathbf{x}) \leq \frac{|\mathcal{Y}| - 1}{|\mathcal{Y}|} \quad j \in \{1, \ldots, |\mathcal{Y}|\}; \quad \sum_{j=1}^{|\mathcal{Y}|} f_j(\mathbf{x}) = 0.$$

The solution for this maximization satisfies $f_j^*(\mathbf{x}) = \frac{|\mathcal{Y}|-1}{|\mathcal{Y}|}$ if $j = \operatorname{argmax}_j P_j(\mathbf{x})$, and $-\frac{1}{|\mathcal{Y}|}$ otherwise. This implies that the adversarial zero-one loss, $\text{AL}^{\text{0-1}}$, from Eq. (4) is Fisher consistent. $\quad\square$

## C   Proof for the Oracle's Greedy Algorithm Optimality (Theorem 4)

**Theorem 4.** *The proposed greedy algorithm used by the oracle is optimal.*

*Proof.* To calculate the set $R$ that maximize $\text{AL}^{\text{0-1}}$ given $\theta$ and a sample $(\mathbf{x}_i, y_i)$, the algorithm calculates all potentials $\psi_{j,y_i}(\mathbf{x}_i)$ for each label $j \in \{1, \ldots, |\mathcal{Y}|\}$ and sorts them from in non-increasing order. Starting with the empty set $R = \emptyset$, it then adds labels to $R$ in sorted order until adding a label would decrease the value of $\frac{\sum_{j \in R} \psi_{j,y_i}(\mathbf{x}_i) + |R| - 1}{|R|}$.

If the set that maximizes $\text{AL}^{\text{0-1}}$ has $k$ elements, it must contain the $k$ largest potentials, otherwise we can swap the potentials that are not in the $k$ largest potentials list with the potentials in the list and get a larger value. We are now left to prove that adding more potentials to the set $R$ will not increase the value of $\frac{\sum_{j \in R} \psi_{j,y_i}(\mathbf{x}_i) + |R| - 1}{|R|}$.

Let $\psi_i$ denote the potentials sorted in non-increasing order, i.e. $\psi_1 \geq \psi_2 \geq \cdots \geq \psi_{|\mathcal{Y}|}$, and let $k$ be the size of the set $R$, hence $\frac{\sum_{j \in R} \psi_{j,y_i}(\mathbf{x}_i) + |R| - 1}{|R|} = \frac{\sum_{i=1}^{k} \psi_i + k - 1}{k}$. We aim to prove that $\frac{\sum_{i=1}^{k} \psi_i + k - 1}{k} \geq \frac{\sum_{i=1}^{k+j} \psi_i + k + j - 1}{k+j}$ for any $j = \{1, \ldots, |\mathcal{Y}| - k\}$. From the construction of the algorithm we know that it is true for $j = 1$, i.e.,:

$$\frac{\sum_{i=1}^{k} \psi_i + k - 1}{k} \geq \frac{\sum_{i=1}^{k+1} \psi_i + k}{k+1} \tag{50}$$

$$(k+1) \left( \sum_{i=1}^{k} \psi_i + k - 1 \right) \geq k \left( \sum_{i=1}^{k+1} \psi_i + k \right) \tag{51}$$

$$k \sum_{i=1}^{k} \psi_i + k^2 - k + \sum_{i=1}^{k} \psi_i + k - 1 \geq k \sum_{i=1}^{k} \psi_i + k\psi_{k+1} + k^2 \tag{52}$$

$$\sum_{i=1}^{k} \psi_i - 1 \geq k\psi_{k+1}. \tag{53}$$

Since the potentials are sorted in non-increasing order, then for any $j = \{1, \ldots, |\mathcal{Y}| - k\}$:

$$j \left( \sum_{i=1}^{k} \psi_i - 1 \right) \geq jk\psi_{k+1} \geq k \sum_{i=1}^{j} \psi_{k+j} \tag{54}$$

$$j\sum_{i=1}^{k}\psi_i - j + k\sum_{i=1}^{k}\psi_i + k^2 + k(j-1) \geq k\sum_{i=1}^{j}\psi_{k+j} + k\sum_{i=1}^{k}\psi_i + k^2 + k(j-1) \tag{55}$$

$$k\sum_{i=1}^{k}\psi_i + k^2 - k + j\sum_{i=1}^{k}\psi_i + jk - j \geq k\sum_{i=1}^{k+j}\psi_i + k^2 + k(j-1) \tag{56}$$

$$(k+j)\left(\sum_{i=1}^{k}\psi_i + k - 1\right) \geq k\left(\sum_{i=1}^{k+j}\psi_i + k + j - 1\right) \tag{57}$$

$$\frac{\sum_{i=1}^{k}\psi_i + k - 1}{k} \geq \frac{\sum_{i=1}^{k+j}\psi_i + k + j - 1}{k+j}. \tag{58}$$

Therefore, we can conclude that the oracle's greedy algorithm is optimal. $\square$

## D  Proof for the Quadratic Programming Formulation (Theorem 5)

**Theorem 5.**  *Let $\Lambda_{i,k}$ be the partial derivative of $\Delta_{i,k}$ with respect to $\theta$, i.e., $\Lambda_{i,k} = \frac{d\Delta_{i,k}}{d\theta}$ and let $\nu_{i,k}$ be the constant part of $\Delta_{i,k}$ (for example if $\Delta_{i,k} = \frac{\psi_{1,y_i}(\mathbf{x}_i) + \psi_{3,y_i}(\mathbf{x}_i) + \psi_{4,y_i}(\mathbf{x}_i) + 2}{3}$, then $\nu_{i,k} = \frac{2}{3}$), then the corresponding dual optimization for the primal minimization (Eq. 5) is:*

$$\max_{\boldsymbol{\alpha}}\sum_{i=1}^{n}\sum_{k=1}^{2^{|\mathcal{Y}|}-1}\nu_{i,k}\,\alpha_{i,k} - \frac{1}{2}\sum_{i,j=1}^{m}\sum_{k,l=1}^{2^{|\mathcal{Y}|}-1}\alpha_{i,k}\alpha_{j,l}\left[\Lambda_{i,k}\cdot\Lambda_{j,l}\right] \tag{59}$$

$$\textit{subject to:}\quad \alpha_{i,k} \geq 0, \quad \sum_{k=1}^{2^{|\mathcal{Y}|}-1}\alpha_{i,k} = C, \; i \in \{1,\dots,n\}, \; k \in \{1,\dots,2^{|\mathcal{Y}|}-1\},$$

*where $\alpha_{i,k}$ is the dual variable for the $k$-th constraint of the $i$-th sample.*

*Proof.*  We can write the Lagrangian for the primal optimization in Eq. 5 as follows:

$$\mathcal{L}(\theta, \boldsymbol{\xi}, \boldsymbol{\alpha}) = \frac{1}{2}\|\theta\|^2 + C\sum_{i=1}^{n}\xi_i - \sum_{i=1}^{n}\alpha_{i,1}[-\Delta_{i,1}+\xi_i] - \cdots - \sum_{i=1}^{n}\alpha_{i,2^{|\mathcal{Y}|}-1}[-\Delta_{i,2^{|\mathcal{Y}|}-1}+\xi_i]. \tag{60}$$

We then write the KKT conditions for optimality and the complementary conditions as follows:

$$\nabla_\theta\mathcal{L} = \theta - \sum_{i=1}^{n}\alpha_{i,1}[-\Lambda_{i,1}] - \cdots - \sum_{i=1}^{n}\alpha_{i,2^{|\mathcal{Y}|}-1}[-\Lambda_{i,2^{|\mathcal{Y}|}-1}] = 0 \quad \Rightarrow \theta = -\sum_{i=1}^{n}\sum_{k=1}^{2^{|\mathcal{Y}|}-1}\alpha_{i,k}\Lambda_{i,k} \tag{61}$$

$$\nabla_{\xi_i}\mathcal{L} = C - \alpha_{i,1} - \cdots - \alpha_{i,2^{|\mathcal{Y}|}-1} = 0 \qquad\qquad \Rightarrow \sum_{k=1}^{2^{|\mathcal{Y}|}-1}\alpha_{i,k} = C \tag{62}$$

$$\forall i,k, \; \alpha_i[-\Delta_{i,k}+\xi_i] = 0 \qquad\qquad\qquad\qquad \Rightarrow \alpha_{i,k} = 0 \vee \xi_i = \Delta_{i,k} \tag{63}$$

Rearranging the Lagrangian formula, and plugging the definition of $\theta$ in terms of dual variables and applying the complementary conditions yields:

$$\mathcal{L} = \frac{1}{2}\|\theta\|^2 + \sum_{i=1}^{n}\alpha_{i,1}[\theta\cdot\Lambda_{i,1}+\nu_{i,1}] + \cdots + \sum_{i=1}^{n}\alpha_{i,2^{|\mathcal{Y}|}-1}[\theta\cdot\Lambda_{i,2^{|\mathcal{Y}|}-1}+\nu_{i,2^{|\mathcal{Y}|}-1}]$$

$$+ \sum_{i=1}^{n}(C - \alpha_{i,1} - \cdots - \alpha_{i,2^{|\mathcal{Y}|}-1})\xi_i \tag{64}$$

$$= \frac{1}{2}\|\theta\|^2 + \sum_{i=1}^{n}\sum_{k=1}^{2^{|\mathcal{Y}|}-1} \alpha_{i,k}\left[\theta \cdot \Lambda_{i,k}\right] + \sum_{i=1}^{n}\sum_{k=1}^{2^{|\mathcal{Y}|}-1} \nu_{i,k}\,\alpha_{i,k} \tag{65}$$

$$= -\frac{1}{2}\sum_{i,j=1}^{n}\sum_{k,l=1}^{2^{|\mathcal{Y}|}-1} \alpha_{i,k}\alpha_{j,l}\left[\Lambda_{i,k}\cdot\Lambda_{j,l}\right] + \sum_{i=1}^{n}\sum_{k=1}^{2^{|\mathcal{Y}|}-1} \nu_{i,k}\,\alpha_{i,k}. \tag{66}$$

Therefore, the dual quadratic programming formulation can be written as:

$$\max_{\boldsymbol{\alpha}} \sum_{i=1}^{n}\sum_{k=1}^{2^{|\mathcal{Y}|}-1} \nu_{i,k}\,\alpha_{i,k} - \frac{1}{2}\sum_{i,j=1}^{m}\sum_{k,l=1}^{2^{|\mathcal{Y}|}-1} \alpha_{i,k}\alpha_{j,l}\left[\Lambda_{i,k}\cdot\Lambda_{j,l}\right] \tag{67}$$

$$\text{subject to} \quad \alpha_{i,k} \geq 0, \quad \sum_{k=1}^{2^{|\mathcal{Y}|}-1} \alpha_{i,k} = C, \; i \in \{1,\dots,n\}, \, k \in \{1,\dots,2^{|\mathcal{Y}|}-1\}.$$

$\square$

# E Proof for the Kernel Trick (Theorem 6)

**Theorem 6.** *Let $\mathcal{X}$ be the input space and $K$ be a positive definite real valued kernel on $\mathcal{X} \times \mathcal{X}$ with a mapping function $\omega(\mathbf{x}) : \mathcal{X} \to \mathcal{H}$ that maps the input space $\mathcal{X}$ to a reproducing kernel Hilbert space $\mathcal{H}$. Then, all the values in the dual optimization of Eq. (6) needed to operate in the Hilbert space $\mathcal{H}$ can be computed in terms of the kernel function $K(\mathbf{x}_i, \mathbf{x}_j)$ as:*

$$\Lambda_{i,k}\cdot\Lambda_{j,l} = c_{(i,k),(j,l)}\, K(\mathbf{x}_i,\mathbf{x}_j), \tag{68}$$

$$\Delta_{i,k} = -\sum_{j=1}^{n}\sum_{l=1}^{2^{|\mathcal{Y}|}-1} \alpha_{j,l}\, c_{(j,l),(i,k)}\, K(\mathbf{x}_j,\mathbf{x}_i) + \nu_{i,k}, \tag{69}$$

$$f_m(\mathbf{x}_i) = -\sum_{j=1}^{n}\sum_{l=1}^{2^{|\mathcal{Y}|}-1} \alpha_{j,l}\left[\left(\frac{\mathbf{1}(m \in R_{j,l})}{|R_{j,l}|} - \mathbf{1}(m = y_j)\right) K(\mathbf{x}_j,\mathbf{x}_i)\right], \tag{70}$$

*where* $c_{(i,k),(j,l)} = \sum_{m=1}^{|\mathcal{Y}|}\left(\dfrac{\mathbf{1}(m \in R_{i,k})}{|R_{i,k}|} - \mathbf{1}(m = y_i)\right)\left(\dfrac{\mathbf{1}(m \in R_{j,l})}{|R_{j,l}|} - \mathbf{1}(m = y_j)\right),$

*and $R_{i,k}$ is the set of labels included in the constraint $\Delta_{i,k}$ (for example if $\Delta_{i,k} = \frac{\psi_{1,y_i}(\mathbf{x}_i)+\psi_{3,y_i}(\mathbf{x}_i)+\psi_{4,y_i}(\mathbf{x}_i)+2}{3}$, then $R_{i,k} = \{1,3,4\}$), the function $\mathbf{1}(j = y_i)$ returns 1 if $j = y_i$ or 0 otherwise, and the function $\mathbf{1}(j \in R_{i,k})$ returns 1 if $j$ is a member of set $R_{i,k}$ or 0 otherwise.*

*Proof.* First, let us define the feature function $\phi(\mathbf{x}_i, j)$ used in our formulation in the input space: $\phi(\mathbf{x}_i, j)$ is a vector containing zeros except for the one corresponding to class $j$, which is equal to $\mathbf{x}_i$. For example, $\phi(\mathbf{x}_i, 1) = [\mathbf{x}_i, \mathbf{0}, \dots, \mathbf{0}]^T$, $\phi(\mathbf{x}_i, 2) = [\mathbf{0}, \mathbf{x}_i, \dots, \mathbf{0}]^T$, and $\phi(\mathbf{x}_i, |\mathcal{Y}|) = [\mathbf{0}, \dots, \mathbf{0}, \mathbf{x}_i]^T$, where $\mathbf{0}$ is a vector containing all zeros with the same length as $\mathbf{x}_i$. Therefore, the vector multiplication $\theta^T\phi(\mathbf{x}_i, j) = \theta_j^T\mathbf{x}_i$, where $\theta_j$ is the vector elements in the parameter space corresponding with class $j$.

By employing kernel methods, the optimization works in the reproducing kernel Hilbert space (RKHS). Therefore, in the kernelized optimization, our feature function is $\phi(\omega(\mathbf{x}_i), j)$. Note that our dual formulation (Eq. 6) depends on the dot product $\Lambda_{i,k}\cdot\Lambda_{j,l}$. By definition, $\Lambda_{i,k} = \frac{1}{|R_{i,k}|}\left(\sum_{j\in R_{i,k}}\left[\phi(\omega(\mathbf{x}_i), j) - \phi(\omega(\mathbf{x}_i), y_i)\right]\right)$. We can expand $\Lambda_{i,k}$ as:

$$\Lambda_{i,k} = \frac{1}{|R_{i,k}|}\sum_{j\in R_{i,k}}\left(\phi(\omega(\mathbf{x}_i), j) - \phi(\omega(\mathbf{x}_i), y_i)\right) \tag{71}$$

$$= \left( \frac{1}{|R_{i,k}|} \sum_{j=1}^{|\mathcal{Y}|} \mathbf{1}(j \in R_{i,k}) \phi(\omega(\mathbf{x}_i), j) \right) - \phi(\omega(\mathbf{x}_i), y_i) \tag{72}$$

$$= \sum_{j=1}^{|\mathcal{Y}|} \left( \frac{\mathbf{1}(j \in R_{i,k})}{|R_{i,k}|} \phi(\omega(\mathbf{x}_i), j) - \mathbf{1}(j = y_i) \phi(\omega(\mathbf{x}_i), j) \right) \tag{73}$$

$$= \sum_{j=1}^{|\mathcal{Y}|} \left( \frac{\mathbf{1}(j \in R_{i,k})}{|R_{i,k}|} - \mathbf{1}(j = y_i) \right) \phi(\omega(\mathbf{x}_i), j). \tag{74}$$

Since $\phi(\omega(\mathbf{x}_i), j)$ is just a vector containing zeros and $\omega(\mathbf{x}_i)$, using the scalar multiplication properties of dot product, we can expand $\Lambda_{i,k} \cdot \Lambda_{j,l}$ as the following:

$$\Lambda_{i,k} \cdot \Lambda_{j,l} = \left[ \sum_{m=1}^{|\mathcal{Y}|} \left( \frac{\mathbf{1}(m \in R_{i,k})}{|R_{i,k}|} - \mathbf{1}(m = y_i) \right) \phi(\omega(\mathbf{x}_i), m) \right]$$
$$\cdot \left[ \sum_{m=1}^{|\mathcal{Y}|} \left( \frac{\mathbf{1}(m \in R_{j,l})}{|R_{j,l}|} - \mathbf{1}(m = y_j) \right) \phi(\omega(\mathbf{x}_j), m) \right] \tag{75}$$

$$= \sum_{m=1}^{|\mathcal{Y}|} \left[ \left( \frac{\mathbf{1}(m \in R_{i,k})}{|R_{i,k}|} - \mathbf{1}(m = y_i) \right) \left( \frac{\mathbf{1}(m \in R_{j,l})}{|R_{j,l}|} - \mathbf{1}(m = y_j) \right) \right.$$
$$\left. (\phi(\omega(\mathbf{x}_i), m) \cdot \phi(\omega(\mathbf{x}_j), m)) \right] \tag{76}$$

$$= \left[ \sum_{m=1}^{|\mathcal{Y}|} \left( \frac{\mathbf{1}(m \in R_{i,k})}{|R_{i,k}|} - \mathbf{1}(m = y_i) \right) \left( \frac{\mathbf{1}(m \in R_{j,l})}{|R_{j,l}|} - \mathbf{1}(m = y_j) \right) \right] [\omega(\mathbf{x}_i) \cdot \omega(\mathbf{x}_j)] \tag{77}$$

$$= \left[ \sum_{m=1}^{|\mathcal{Y}|} \left( \frac{\mathbf{1}(m \in R_{i,k})}{|R_{i,k}|} - \mathbf{1}(m = y_i) \right) \left( \frac{\mathbf{1}(m \in R_{j,l})}{|R_{j,l}|} - \mathbf{1}(m = y_j) \right) \right] K(\mathbf{x}_i, \mathbf{x}_j). \tag{78}$$

Let us define:

$$c_{(i,k),(j,l)} = \sum_{m=1}^{|\mathcal{Y}|} \left( \frac{\mathbf{1}(m \in R_{i,k})}{|R_{i,k}|} - \mathbf{1}(m = y_i) \right) \left( \frac{\mathbf{1}(m \in R_{j,l})}{|R_{j,l}|} - \mathbf{1}(m = y_j) \right), \tag{79}$$

then, we have $\Lambda_{i,k} \cdot \Lambda_{j,l} = c_{(i,k),(j,l)} K(\mathbf{x}_i, \mathbf{x}_j)$. We can also express $\Delta_{i,k}$ in terms of kernel functions as the following:

$$\Delta_{i,k} = \theta \cdot \Lambda_{i,k} + \nu_{i,k} = -\sum_{j=1}^{n} \sum_{l=1}^{2^{|\mathcal{Y}|}-1} \alpha_{j,l} \left[ \Lambda_{j,l} \cdot \Lambda_{i,k} \right] + \nu_{i,k} \tag{80}$$

$$= -\sum_{j=1}^{n} \sum_{l=1}^{2^{|\mathcal{Y}|}-1} \alpha_{j,l} \, c_{(j,l),(i,k)} \, K(\mathbf{x}_j, \mathbf{x}_i) + \nu_{i,k}. \tag{81}$$

In the prediction step, given a new datapoint $\mathbf{x}_i$, we need to calculate $f_m(\mathbf{x}_i) = \theta^T \phi(\mathbf{x}_i, m)$ for all $m \in \{1, \ldots, |\mathcal{Y}|\}$. We can also compute $f_m(\mathbf{x}_i)$ in terms of kernel functions as the following:

$$f_m(\mathbf{x}_i) = \theta \cdot \phi(\omega(\mathbf{x}_i), m) \tag{82}$$

$$= -\sum_{j=1}^{n} \sum_{l=1}^{2^{|\mathcal{Y}|}-1} \alpha_{j,l} \left[ \Lambda_{j,l} \cdot \phi(\omega(\mathbf{x}_i), m) \right] \tag{83}$$

$$= -\sum_{j=1}^{n} \sum_{l=1}^{2^{|\mathcal{Y}|}-1} \alpha_{j,l} \left[ \left( \sum_{q=1}^{|\mathcal{Y}|} \left( \frac{\mathbf{1}(q \in R_{j,l})}{|R_{j,l}|} - \mathbf{1}(q = y_j) \right) \phi(\omega(\mathbf{x}_j), q) \right) \cdot \phi(\omega(\mathbf{x}_i), m) \right] \tag{84}$$

$$= -\sum_{j=1}^{n} \sum_{l=1}^{2^{|\mathcal{Y}|}-1} \alpha_{j,l} \left[ \left( \frac{\mathbf{1}(m \in R_{j,l})}{|R_{j,l}|} - \mathbf{1}(m = y_j) \right) \phi(\omega(\mathbf{x}_j), m) \cdot \phi(\omega(\mathbf{x}_i), m) \right] \tag{85}$$

$$= -\sum_{j=1}^{n} \sum_{l=1}^{2^{|\mathcal{Y}|}-1} \alpha_{j,l} \left[ \left( \frac{\mathbf{1}(m \in R_{j,l})}{|R_{j,l}|} - \mathbf{1}(m = y_j) \right) \omega(\mathbf{x}_j) \cdot \omega(\mathbf{x}_i) \right] \tag{86}$$

$$= -\sum_{j=1}^{n} \sum_{l=1}^{2^{|\mathcal{Y}|}-1} \alpha_{j,l} \left[ \left( \frac{\mathbf{1}(m \in R_{j,l})}{|R_{j,l}|} - \mathbf{1}(m = y_j) \right) K(\mathbf{x}_j, \mathbf{x}_i) \right]. \tag{87}$$

$\square$

## F  Proof for the Polynomial Convergence Analysis (Theorem 7)

**Theorem 7.** *For any $\epsilon > 0$ and training dataset $\{(\mathbf{x}_1, y_1), \ldots, (\mathbf{x}_n, y_n)\}$ with $U = \max_i [\mathbf{x}_i \cdot \mathbf{x}_i]$, Algorithm 1 terminates after incrementally adding at most $\max\left\{ \frac{2n}{\epsilon}, \frac{4nCU}{\epsilon^2} \right\}$ constraints to the constraint set $A^*$.*

*Proof.* First, we want to establish a lower bound on the improvement of the dual objective value after each time we add an additional constraint. The proof follows the structure of the proof described by Tsochantaridis et al. [27].

For the purpose of simplification, let us change the index of the dual QP variable. Assuming a combined index $s = (i, k)$ represents the $i$-th sample and $k$-th constraint, we can rewrite our dual QP objective as:

$$W(\boldsymbol{\alpha}) = \sum_s \nu_s \, \alpha_s - \frac{1}{2} \sum_{s,t} \alpha_s \alpha_t [\Lambda_s \cdot \Lambda_t]. \tag{88}$$

In the constraint generation steps, we only consider the constraints that are already added to the constraint set $A^*$. In the simplification above, we can set $\alpha_s = 0$ for all constraints that are not yet added to the set. When we add a new constraint $\xi_i \geq \Delta_s$, we allow $\alpha_s$ to have value from 0 to $C$, but it needs to maintain the constraint that $\sum_{k=1}^{2^{|\mathcal{Y}|}-1} \alpha_{i,k} = C$. Therefore, we cannot analyze the improvement in the dual objective for adding one constraint by just optimizing over $\alpha_s$ alone. We need to consider a larger optimization over the whole space.

For the purpose of deriving bounds, it is sufficient to restrict our attention to a one-dimensional version of the optimization, i.e., trying to optimize in just one specific direction. If we can show that it makes sufficient improvement on just one specific direction, it implies that the optimization over the whole space can improve at least that much on the objective function.

Let us consider adding one new constraint where we allow $\alpha_r$ taking values other than 0. Let $\beta$ be the value we set for $\alpha_r$ and $W'(\boldsymbol{\alpha})$ be the new objective value after we add the new constraint. Then the difference of the objective value between before and after adding the constraint is:

$$W'(\boldsymbol{\alpha}) - W(\boldsymbol{\alpha}) = \nu_r \beta - \frac{1}{2} \left( 2 \sum_s \alpha_s \beta [\Lambda_s \cdot \Lambda_r] + \beta^2 [\Lambda_r \cdot \Lambda_r] \right) \tag{89}$$

$$= \beta \left( \nu_r - \sum_s \alpha_s [\Lambda_s \cdot \Lambda_r] \right) - \frac{\beta^2}{2} [\Lambda_r \cdot \Lambda_r]. \tag{90}$$

The optimization needs to find $\beta^*$ that maximizes the formula above. We can find $\beta^*$ by taking the derivative of the difference above with respect to $\beta$ and setting it to zero.

$$\frac{d(W'(\boldsymbol{\alpha}) - W(\boldsymbol{\alpha}))}{d\beta} = \nu_r - \sum_s \alpha_s [\Lambda_s \cdot \Lambda_r] - \beta [\Lambda_r \cdot \Lambda_r] = 0 \tag{91}$$

$$\beta^* = \frac{\nu_r - \sum_s \alpha_s [\Lambda_s \cdot \Lambda_r]}{\Lambda_r \cdot \Lambda_r} = \frac{\nu_r + \theta \cdot \Lambda_r}{\Lambda_r \cdot \Lambda_r} = \frac{\Delta_r}{\Lambda_r \cdot \Lambda_r}. \tag{92}$$

Note that $\beta^* > 0$ because $[\Lambda_r \cdot \Lambda_r] > 0$ and the algorithm only add one constraint where $\Delta_r > 0$. The improvement of the dual objective value after adding the new constraint can be computed as follows:

$$\max_{\beta > 0} [W'(\boldsymbol{\alpha}) - W(\boldsymbol{\alpha})] = \frac{(\Delta_r)^2}{[\Lambda_r \cdot \Lambda_r]} - \frac{(\Delta_r)^2}{2([\Lambda_r \cdot \Lambda_r])^2}[\Lambda_r \cdot \Lambda_r] = \frac{(\Delta_r)^2}{2[\Lambda_r \cdot \Lambda_r]}. \tag{93}$$

Note that this improvement is always positive.

Let us denote the index $r$ as an index consisting of the pair $(i, k)$. By the definition, $\Lambda_r = \frac{1}{|R_r|}\left(\sum_{j \in R_r}[\phi(\mathbf{x}_i, j) - \phi(\mathbf{x}_i, y_i)]\right)$, where $R_r$ is the set of classes included in the constraint $\Delta_r$. Using Equation (78), we analyze the dot product $\Lambda_r \cdot \Lambda_r$ as follows:

$$\Lambda_r \cdot \Lambda_r = \left[\sum_{m=1}^{|\mathcal{Y}|}\left(\frac{\mathbf{1}(m \in R_r)}{|R_r|} - \mathbf{1}(m = y_i)\right)\left(\frac{\mathbf{1}(m \in R_r)}{|R_r|} - \mathbf{1}(m = y_i)\right)\right][\mathbf{x}_i \cdot \mathbf{x}_i] \tag{94}$$

$$= \frac{1}{|R_r|^2}\left[\sum_{m=1}^{|\mathcal{Y}|}[\mathbf{1}(m \in R_r) - |R_r|\mathbf{1}(m = y_i)][\mathbf{1}(m \in R_r) - |R_r|\mathbf{1}(m = y_i)]\right][\mathbf{x}_i \cdot \mathbf{x}_i] \tag{95}$$

$$\leq \frac{|R_r|^2 + |R_r|}{|R_r|^2}[\mathbf{x}_i \cdot \mathbf{x}_i] = \left(1 + \frac{1}{|R_r|}\right)[\mathbf{x}_i \cdot \mathbf{x}_i] \leq 2[\mathbf{x}_i \cdot \mathbf{x}_i]. \tag{96}$$

Let $U$ be the maximum of $[\mathbf{x}_i \cdot \mathbf{x}_i]$ over all training data, i.e., $U = \max_i[\mathbf{x}_i \cdot \mathbf{x}_i]$. Plugging the result above into the improvement, we have:

$$W'(\boldsymbol{\alpha}) - W(\boldsymbol{\alpha}) = \frac{(\Delta_r)^2}{2[\Lambda_r \cdot \Lambda_r]} \geq \frac{(\Delta_r)^2}{4[\mathbf{x}_i \cdot \mathbf{x}_i]} \geq \frac{(\Delta_r)^2}{4U}. \tag{97}$$

Note that there is also a restriction: $\alpha_r \leq C$. In the case of $\beta^* > C$, we need to adjust the improvement. If $\beta^* > C$, it also implies that:

$$\frac{\Delta_r}{\Lambda_r \cdot \Lambda_r} > C \quad \Leftrightarrow \quad \Delta_r > C[\Lambda_r \cdot \Lambda_r]. \tag{98}$$

Therefore, the improvement of the dual objective after adding the constraint with the restriction $0 < \beta \leq C$ is:

$$\max_{0 < \beta \leq C}[W'(\boldsymbol{\alpha}) - W(\boldsymbol{\alpha})] = \begin{cases} \frac{(\Delta_r)^2}{2[\Lambda_r \cdot \Lambda_r]} & \text{if } \Delta_r \leq C[\Lambda_r \cdot \Lambda_r] \\ C\Delta_r - \frac{C^2}{2}[\Lambda_r \cdot \Lambda_r] & \text{otherwise} \end{cases} \tag{99}$$

$$\geq \frac{\Delta_r}{2}\min\left\{C, \frac{\Delta_r}{[\Lambda_r \cdot \Lambda_r]}\right\} \geq \frac{\Delta_r}{2}\min\left\{C, \frac{\Delta_r}{2U}\right\}. \tag{100}$$

Note that the dual objective value is upper-bounded by $nC$, where $n$ is the number of training data. Since in every iteration the algorithm will add a new constraint if $\Delta_r \geq \xi_i + \epsilon$ and the objective get the improvement as described above for each additional constraint, the algorithm will terminate after incrementally adding a number of constraints that is at most:

$$\max\left\{\frac{2nC}{\Delta_r C}, \frac{4nCU}{(\Delta_r)^2}\right\} \leq \max\left\{\frac{2n}{\epsilon}, \frac{4nCU}{\epsilon^2}\right\}. \tag{101}$$

$\square$

# G  Proof of the Matrix Inverse in the Equilibrium Game Value Analysis

**Lemma 5.** *The matrix $M^{-1}$ in Eq. 10 is the inverse of the matrix $M$ in Eq. 9, i.e.*

$$MM^{-1} = I$$

*Proof.* Let us denote $H = MM^{-1}$. We want to prove that $H = I$. We will prove the equality by analyzing each cell $H_{k,l}$ of the matrix. The value of $H_{k,l}$ should be 1 if $k$ is equal to $l$ and 0 otherwise.

For the $k$-th diagonal entry in the matrix $H$, we have:

$$H_{k,k} = -\frac{\psi_{k,y_i}(\mathbf{x}_i)\left(\sum_{j=1,j\neq k}^{|\mathcal{Y}|}\psi_{j,y_i}(\mathbf{x}_i)+|\mathcal{Y}|-2\right)}{\sum_{j=1}^{|\mathcal{Y}|}\psi_{j,y_i}(\mathbf{x}_i)+|\mathcal{Y}|-1} + \frac{\sum_{j=1,j\neq k}^{|\mathcal{Y}|}\left(\psi_{k,y_i}(\mathbf{x}_i)+1\right)\left(\psi_{j,y_i}(\mathbf{x}_i)+1\right)}{\sum_{j=1}^{|\mathcal{Y}|}\psi_{j,y_i}(\mathbf{x}_i)+|\mathcal{Y}|-1}$$

$$\tag{102}$$

$$= -\frac{\psi_{k,y_i}(\mathbf{x}_i)\sum_{j=1,j\neq k}^{|\mathcal{Y}|}\psi_{j,y_i}(\mathbf{x}_i)+(|\mathcal{Y}|-2)\psi_{k,y_i}(\mathbf{x}_i)}{\sum_{j=1}^{|\mathcal{Y}|}\psi_{j,y_i}(\mathbf{x}_i)+|\mathcal{Y}|-1}$$

$$+\frac{\psi_{k,y_i}(\mathbf{x}_i)\sum_{j=1,j\neq k}^{|\mathcal{Y}|}\psi_{j,y_i}(\mathbf{x}_i)+\sum_{j=1,j\neq k}^{|\mathcal{Y}|}\psi_{j,y_i}(\mathbf{x}_i)+(|\mathcal{Y}|-1)\psi_{k,y_i}(\mathbf{x}_i)+|\mathcal{Y}|-1}{\sum_{j=1}^{|\mathcal{Y}|}\psi_{j,y_i}(\mathbf{x}_i)+|\mathcal{Y}|-1}$$

$$\tag{103}$$

$$=\frac{\sum_{j=1,j\neq k}^{|\mathcal{Y}|}\psi_{j,y_i}(\mathbf{x}_i)+\psi_{k,y_i}(\mathbf{x}_i)+|\mathcal{Y}|-1}{\sum_{j=1}^{|\mathcal{Y}|}\psi_{j,y_i}(\mathbf{x}_i)+|\mathcal{Y}|-1}$$

$$\tag{104}$$

$$=\frac{\sum_{j=1}^{|\mathcal{Y}|}\psi_{j,y_i}(\mathbf{x}_i)+|\mathcal{Y}|-1}{\sum_{j=1}^{|\mathcal{Y}|}\psi_{j,y_i}(\mathbf{x}_i)+|\mathcal{Y}|-1}$$

$$\tag{105}$$

$$=1,\tag{106}$$

and for non-diagonal entries $H_{k,l}$, where $k \neq l$, we have:

$$H_{k,l} = -\frac{(\psi_{k,y_i}(\mathbf{x}_i)+1)\left(\sum_{j=1,j\neq l}^{|\mathcal{Y}|}\psi_{j,y_i}(\mathbf{x}_i)+|\mathcal{Y}|-2\right)}{\sum_{j=1}^{|\mathcal{Y}|}\psi_{j,y_i}(\mathbf{x}_i)+|\mathcal{Y}|-1} + \frac{\psi_{k,y_i}(\mathbf{x}_i)\left(\psi_{k,y_i}(\mathbf{x}_i)+1\right)}{\sum_{j=1}^{|\mathcal{Y}|}\psi_{j,y_i}(\mathbf{x}_i)+|\mathcal{Y}|-1}$$

$$+\frac{\sum_{j=1,j\neq k,j\neq l}^{|\mathcal{Y}|}\left(\psi_{k,y_i}(\mathbf{x}_i)+1\right)\left(\psi_{j,y_i}(\mathbf{x}_i)+1\right)}{\sum_{j=1}^{|\mathcal{Y}|}\psi_{j,y_i}(\mathbf{x}_i)+|\mathcal{Y}|-1}$$

$$\tag{107}$$

$$= -\frac{\psi_{k,y_i}(\mathbf{x}_i)\sum_{j=1,j\neq l}^{|\mathcal{Y}|}\psi_{j,y_i}(\mathbf{x}_i)+(|\mathcal{Y}|-2)\psi_{k,y_i}(\mathbf{x}_i)+\sum_{j=1,j\neq l}^{|\mathcal{Y}|}\psi_{j,y_i}(\mathbf{x}_i)+|\mathcal{Y}|-2}{\sum_{j=1}^{|\mathcal{Y}|}\psi_{j,y_i}(\mathbf{x}_i)+|\mathcal{Y}|-1}$$

$$+\frac{\psi_{k,y_i}(\mathbf{x}_i)\left(\psi_{k,y_i}(\mathbf{x}_i)+1\right)}{\sum_{j=1}^{|\mathcal{Y}|}\psi_{j,y_i}(\mathbf{x}_i)+|\mathcal{Y}|-1}$$

$$+\frac{\psi_{k,y_i}(\mathbf{x}_i)\sum_{\substack{j=1,\\j\neq k,\\j\neq l}}^{|\mathcal{Y}|}\psi_{j,y_i}(\mathbf{x}_i)+\sum_{\substack{j=1,\\j\neq k,\\j\neq l}}^{|\mathcal{Y}|}\psi_{j,y_i}(\mathbf{x}_i)+(|\mathcal{Y}|-2)\psi_{k,y_i}(\mathbf{x}_i)+|\mathcal{Y}|-2}{\sum_{j=1}^{|\mathcal{Y}|}\psi_{j,y_i}(\mathbf{x}_i)+|\mathcal{Y}|-1}$$

$$\tag{108}$$

$$= -\frac{\psi_{k,y_i}(\mathbf{x}_i)\sum_{j=1,j\neq l}^{|\mathcal{Y}|}\psi_{j,y_i}(\mathbf{x}_i)+\sum_{j=1,j\neq l}^{|\mathcal{Y}|}\psi_{j,y_i}(\mathbf{x}_i)}{\sum_{j=1}^{|\mathcal{Y}|}\psi_{j,y_i}(\mathbf{x}_i)+|\mathcal{Y}|-1}$$

$$+\frac{\psi_{k,y_i}(\mathbf{x}_i)\sum_{j=1,j\neq l}^{|\mathcal{Y}|}\psi_{j,y_i}(\mathbf{x}_i)+\sum_{j=1,j\neq l}^{|\mathcal{Y}|}\psi_{j,y_i}(\mathbf{x}_i)}{\sum_{j=1}^{|\mathcal{Y}|}\psi_{j,y_i}(\mathbf{x}_i)+|\mathcal{Y}|-1}$$

$$\tag{109}$$

$$=0\tag{110}$$

Therefore, since all diagonal entries of $H$ has value 1, and all non-diagonal entries of $H$ has zero value, then $MM^{-1} = I$. □

## H Illustrations for the Binary Classification Cases

In the binary classification case, the adversarial loss is the maximum among three functions: $\psi_{1,y_i}(\mathbf{x}_i)$, $\psi_{2,y_i}(\mathbf{x}_i)$, and $\frac{\psi_{1,y_i}(\mathbf{x}_i)+\psi_{2,y_i}(\mathbf{x}_i)+1}{2}$. In the case where the true label $y_i = 1$, we have $\psi_{1,y_i}(\mathbf{x}_i) = 0$. The adversarial loss in this case can be computed as $\max\{0, \psi_{2,1}(\mathbf{x}_i), \frac{\psi_{2,1}(\mathbf{x}_i)+1}{2}\}$, which has values:

$$\text{AL}^{0\text{-}1}_{\text{binary}}|_{y_i=1} = \begin{cases} 0 & \text{if } \psi_{2,1}(\mathbf{x}_i) \leq -1 \\ \psi_{2,1}(\mathbf{x}_i) & \text{if } \psi_{2,1}(\mathbf{x}_i) \geq 1 \\ \frac{\psi_{2,1}(\mathbf{x}_i)+1}{2} & \text{if } -1 \leq \psi_{2,1}(\mathbf{x}_i) \leq 1. \end{cases} \tag{111}$$

Note that $\psi_{2,1}(\mathbf{x}_i) = f_2(\mathbf{x}_i) - f_1(\mathbf{x}_i) = \theta^{\mathrm{T}}\phi(\mathbf{x}_i, 2) - \theta^{\mathrm{T}}\phi(\mathbf{x}_i, 1)$, where $\phi(\mathbf{x}_i, j)$ is a vector containing zero elements except the one corresponding to class $j$ which is equal to $\mathbf{x}_i$. If we change our notation for the class label from $y \in \{1, 2\}$ to $y \in \{-1, +1\}$ and define the parameter $\theta$ to contains both vector parameter $\mathbf{w}$ and bias $b$, the binary adversarial loss can be equivalently formulated as:

$$\text{AL}^{0\text{-}1}_{\text{binary}} = \begin{cases} 0 & \text{if } y_i(\mathbf{w} \cdot \mathbf{x}_i + b) \geq 1 \\ \frac{-y_i(\mathbf{w}\cdot\mathbf{x}_i+b)+1}{2} & \text{if } -1 \leq y_i(\mathbf{w} \cdot \mathbf{x}_i + b) \leq 1 \\ -y_i(\mathbf{w} \cdot \mathbf{x}_i + b) & \text{if } y_i(\mathbf{w} \cdot \mathbf{x}_i + b) \leq -1. \end{cases} \tag{112}$$

Adding L2 regularization to the binary adversarial loss and introducing slack variables $\xi_i$ and $\delta_i$ results in the following quadratic programming formulation:

$$\min_{\mathbf{w},b} \quad \frac{1}{2}\|\mathbf{w}\|^2 + C\left[\sum_{i=1}^{m}\frac{1}{2}\xi_i + \sum_{i=1}^{m}\frac{1}{2}\delta_i\right] \tag{113}$$
$$\text{subject to} \quad y_i(\mathbf{w} \cdot \mathbf{x}_i + b) \geq 1 - \xi_i$$
$$y_i(\mathbf{w} \cdot \mathbf{x}_i + b) \geq -1 - \delta_i$$
$$\xi_i \geq 0$$
$$\delta_i \geq 0$$
$$i \in \{1, \ldots, m\}.$$

Note that the formulation above is similar to the formulation of SVMs. The difference is that the adversarial formulation has two slack variables corresponding to the hinges at 1 and -1.

We can view the adversarial formulation as maximizing a margin that is similar to the soft-margin SVM, but with different constraints. We study how this adversarial formulation's double hinges affect the maximum margin in its solutions. Figure 5 shows the comparison of the maximum margin resulted from the adversarial method and SVM for different values of the $C$.

As we can see from the figure, the adversarial solution tends to have larger margins than the SVM solution under identical choices of $C$. In the case where $C = 10$ and $C = 100$, the adversarial solution is very similar to the SVM solution, with different choice of the support vector points that define the margin.

The interesting results can be seen in the case where $C = 1000$ and $C = 10000$. In the SVM solution, the marginal hyperplanes (i.e., the line $\mathbf{w} \cdot \mathbf{x}_i + b = \pm 1$) that define the boundary of the margin always cross some support vectors that are classified correctly by the algorithm (highlighted with red in the figure). In the adversarial solution, however, the marginal hyperplanes may also cross some support vectors that are classified incorrectly by the algorithm (highlighted with green in the figure). For example in the case where $C = 10000$, the marginal hyperplanes in the adversarial solution are defined by three support vectors, one of them is classified correctly and two of them are classified incorrectly. This kind of solution is unique to the adversarial method with no possibility of being realized under the standard SVM algorithm.

Figure 5: The maximum margin hyperplanes of the adversarial classification and SVM for different values of $C$.

## Footnotes

[3]Emmanual N Barron. Game Theory: An Introduction, Volume 2. John Wiley & Sons, 2013.