[Reviews · NeurIPS 2016]

Reviewer 1

Summary

The authors propose a consistent loss function for multi-class SVM based on an adversarial formulation. They provide several interesting geometric comparisons with popular losses for multi-class SVM. They prove the consistency of their loss and show that it also has competitive performance when compared against popular multi-class SVM formulations such as the Crammer-Singer and Weston-Watkins.

Qualitative Assessment

- Overall I enjoy reading this paper. The authors derive a new consistent loss function for multi-class SVM through adversarial classification and show geometrically how the loss function differ from others. It is the first consistent loss based on relative margin. - The experiments show that with the use of relative margin instead of absolute margin, the proposed method is better than the previous consistent approach LLW on low dimensional datasets. - The transition from Eqt 2 to Eqt 3 is not explained; it should be clarified. - What is the runtime of the proposed algorithm, when compared to existing algorithms such as WW and CS? This could be an important criterion when choosing algorithms. - The column of SVM constraints in Table 1 is slightly misleading because those are the worst case numbers. In practice few of will be active since many of the problems are reasonably separable and have low loss.

Confidence in this Review

2-Confident (read it all; understood it all reasonably well)


Reviewer 2

Summary

This paper studies multi-class classification. Based on adversarial game formulation proposed in [16], the authors show that the adversarial game formulation is equivalent to an empirical risk minimisation where the loss function is a point wise maximum of 2^{|Y|} cost functions (|Y| is the number of classes). The authors then proved that this loss is Fisher consistent, and archives comparable empirical results to existing methods that are not Fisher consistent, and significantly outperforms the existing consistent method.

Qualitative Assessment

This paper studies multi-class classification. Based on adversarial game formulation proposed in [16], the authors show that the adversarial game formulation is equivalent to an empirical risk minimisation where the loss function is a point wise maximum of 2^{|Y|} cost functions (|Y| is the number of classes). The authors then proved that this loss is Fisher consistent, and archives comparable empirical results to existing methods that are not Fisher consistent, and significantly outperforms the existing consistent method. My main concern is whether the contribution is significant enough. The main result appears to be relating the adversarial game formulation proposed in [16] to the ERM framework (of a rather complicated loss function), and it does appear incremental to me. A second issue I have is that unlike ERM, the “adversarial game” formulation has not been a standard scheme in the machine learning field yet, and more effort may be useful to convince the readers (this reviewer at least) that this is a well motivated framework as opposed to being ad hoc. Below are some more detailed comments: 1. From time to time, I feel the exposition of the paper is not very self-contained. For example, in Theorem 1, the authors mentioned that the “multi class zero-one adversarial classification are equivalent…”. Presumably this is about Formulation (2). However, Formulation (2) is not precisely defined - the function \phi(,) in (2) is not defined, except mentioned in the pass that “(it) prove feature moments measured from sample training data…”Maybe the authors expect the readers are familiar with [16]? 2. I find Theorem 2 hard to parse. Theorem 2 says “The loss for… resides on the hyperplane…”. The loss is a scalar, and what does it mean by a scalar resides on a hyperplane? 3. I think a brief discussion about kernelization suffices. 4. L188: The author mentions that the algorithm ends with a solution violating no constraint by more than \epsilon - I wonder how this near-feasibility guarantee transforms to a guarantee of the quality of the solution to the learning task itself. 5. L189: “polynomial run time convergence bound”. I think it should be pseudo-polynomial, since it is polynomial to \epsilon, not to \log\epsilon.

Confidence in this Review

2-Confident (read it all; understood it all reasonably well)


Reviewer 3

Summary

The paper addresses a multi-class classification problem formulated as a constrained adversarial minimax zero-sum game. The paper builds on [6] where the minimax formulation was first proposed and solved via sub-gradient methods which involve solving many LP's in order to evaluate the sub-gradient. In contrast, the authors of the paper under review show that the minimax game can be reformulated as an empirical risk minimization problem with a particular loss function, termed adversarial zero-one loss (AL01). The resulting ERM problem is equivalent to a convex quadratic program (provided a quadratic regularization is added) with a number of constraints growing exponential with the number of classes. To solve the QP, the authors propose a variant of column generation algorithm [28]. The authors show that the proposed AL01 is Fisher consistent. Moreover, experimental comparison against common multi-class SVM surrogates suggest that the proposed AL01 is not only theoretically sound but also works consistently well in practice.

Qualitative Assessment

It is very interesting paper. Only the presentation should be improved: - The Section 2 is not selfcontained and hard to read due to careless problem formulation and usage of undefined notation, e.g. missing definition of most variables in (2) and (3), or undefined mapping $\phi(x,j)$ that is in fact composed of appropriately stacked vectors $x$ and zero vectors etc. The section implicitly assumes that the reader already knows the MinMax formulation of [6]. - It should be indicated by appropriate notation that the function $\psi_{j,y_i}(x)$ depends on the parameters $\theta$ to be learned, e.g. $\psi_{j,y_i)(x,\theta)$. - Presumably, the computation time might be a bottleneck of the proposed method as the QP has exponential number of constraints in contrast to other multi-class SVM problems. Hence, at least a rough runtime comparison for the evaluated methods should be reported. - In proof of Lemma 1 (supplementary), it is not clear where the formulas for explicitly computing the matrix inverse $M^{-1}$ come from.

Confidence in this Review

2-Confident (read it all; understood it all reasonably well)


Reviewer 4

Summary

The paper unifies adversarial 0/1 loss minimization and empirical risk minimization via a novel loss function. They also propose a quadratic program formulation and an efficient optimization procedure. The resulting learning machine is both Fisher consistent and universally consistent. In addition, several experiments support the validity of the claims.

Qualitative Assessment

This is a nice work. To the best of my knowledge, apart from the Lee Lin and Wahba SVM, this is the first multiclass kernel machine enjoying both Fisher consistency and universal consistency. The connection between the adversarial 0/1 loss and ERM may potentially have an impact beyond the problem tackled in this paper. The relevant literature is well cited The algorithmic contribution is not earth-shattering or novel. Although I like the constrained generation method (proposed in algorithm 1) to sidestep the quadratic dependency on the number of classes, a similar method has already been used in Wang et al. 2015 (Multivariate prediction Games). I am a bit disappointed by the experiments conducted to assess the effectiveness of the proposed approach. Only very small datasets are used. LLW is known to perform poorly on these datasets (small feature size). The same picture does not hold once the dimensionality increases. Experiments on much larger datasets would have been a plus.

Confidence in this Review

2-Confident (read it all; understood it all reasonably well)


Reviewer 5

Summary

This is a very interesting and well-written paper that closes the gap between theory and practice by providing a Fisher- and universally consistent loss function for multiclass classification that performs empirically well in practice. The authors show how adversarial loss minimization in the multiclass setting can be recast into an empirical risk minimization setup under the adversarial 0-1 loss function. They further prove that the adversarial 0-1 loss is both Fisher and universally consistent (so that it has advantageous theoretical properties) and demonstrate a novel algorithm that for efficient optimization that uses a constraint generation QP method (allowing the method to be computationally efficient). The experiments are detailed and thorough, testing on 12 UCI datasets and comparing the adversarial 0-1 loss to three other multiclass SVM methods.

Qualitative Assessment

The main theorems of the paper are technically rich, and support the conclusion of the paper. The proofs are extremely well documented in the supplementary material, and easy to follow proof sketches are given in the main body of the work. The experimental work compares the proposed method to three other competing state of the art multiclass SVM solvers across 12 datasets. The statistical significance of the experimental results are considered; 20 dataset splits are created and a paired t-test is performed. Additionally, the results are convincing in that they agree with previous work, as explained in lines 233-234. However, I am curious as to why the authors did not compare against OVA multiclass classification since it has been extremely popular for its computational efficiency and shown to perform as well statistically as other methods (see Rifkin “In defense of One vs. All.”) One main weakness of the experimental results is that they claim that the new algorithm is computationally efficient and provide a proof of convergence, but do not include any performance vs. time results that back up these claims. Novelty/Originality The work is highly novel in that it proposes a new loss function and a new, efficient algorithm to solve the associated optimization problem. Potential impact or usefulness The impact of the work is high since it combines work done from two somewhat distinct areas (adversarial classification and consistency of empirical risk minimization methods). In addition, the SVM algorithm for multiclass is highly popular, so I expect this result to affect a large number of people. Clarity and presentation Overall, the paper is well written. The intro very clearly lays out the gaps in existing approaches, and explains how the AL-01 approach bridges these gaps. The background gives a clear introduction to the main ideas needed to understand the theoretical results of the paper. The explanations in the experimental section make the findings easy to understand. Some specific comments… Figure 1 and Figure 2. These are difficult to understand, but I feel could be potentially very elucidating. Spend a little more time in text going over how to interpret these figures? Table 2 #b - not clear what this means immediately. Should explain in Figure caption.

Confidence in this Review

1-Less confident (might not have understood significant parts)


Reviewer 6

Summary

The paper discusses a mini-max approach to multi-class classification opposed to ERM approach. They consider a recently proposed model by Asif et al.[16], where the empirical training data is replaced by an adversarialy chosen conditional label distribution. The paper extends the results in [16] in the following manner: -- They devise new loss function (AL01) and show that the parameters for [16] can be obtained by empirical risk minimization w.r.t to AL01. -- They prove Fisher Consistency for the approach. -- They develop a quadratic program for extracting the model parameters: This saves computation of a linear program for each subgradient computation. The authors consider as main contribution: Developing a fisher consistent loss function that performs well in practice.

Qualitative Assessment

The paper indeed demonstrates consistency of the aforementioned new loss function. Considering Thm 2: It seems that consistency follows from the fact that in the limit AL01 becomes LLW: a loss function that is criticized as not performing well in practice. It seems the practical advantage of AL01 stems from its difference from LLW in the non-limit: This difference and how it improves LLW is not thoroughly discussed. The paper can also be improved by adding a more thorough discussion and rigorous analysis on the improvement in terms of optimization w.r.t to [16]. To summarize: Demonstrating that [16] is equivalent to some loss function is a nice interesting result, that might deserve further investigation. I am not sure that demonstrating consistency for the method is a significant contribution: in particular because it follows from equivalence to LLW. There is lack of discussion on the improvement w.r.t LLW over finite sample size. Further rigorous analysis of the improvement over [16] in terms of optimization can be helpful for the reader.

Confidence in this Review

2-Confident (read it all; understood it all reasonably well)